
**Characterisation of the filter inlet system on the BAE-146 research aircraft and its use for size**
**resolved aerosol composition measurements**
Alberto Sanchez-Marroquin[1], Duncan H. P. Hedges[1], Matthew Hiscock[2], Simon T. Parker[3], Philip D. Rosenberg[1], Jamie
Trembath[4], Richard Walshaw[1], Ian T. Burke[1], James B. McQuaid[1], Benjamin J. Murray[1]
[1]School of Earth and Environment, University of Leeds, Woodhouse Lane, Leeds, LS2 9JT, UK
[2] Oxford Instruments NanoAnalysis, High Wycombe, HP12 3SE
[3]Defence Science and Technology Laboratory, Salisbury, SP4 0JQ, UK
[4]Facility for Airborne Atmospheric Measurements, Building 146, Cranfield University, College Road, Cranfield, Bedford
MK43 0AL
**Abstract**
Atmospheric aerosol particles are important for our planet's climate because they interact with
radiation and clouds. Hence, having characterised methods to collect aerosol from aircraft for detailed
offline analysis are valuable. However, collecting aerosol, particularly coarse mode aerosol, onto
substrates from a fast moving aircraft is challenging and can result in both losses and enhancement in
aerosol. Here we present the characterisation of an inlet system designed for collection of aerosol
onto filters on board the UK's BAe 146 Facility for Airborne Atmospheric Measurements (FAAM)
research aircraft. We also present an offline Scanning Electron Microscopy (SEM) technique for
quantifying both the size distribution and size resolved composition of the collected aerosol. We use
this SEM technique in parallel with online underwing optical probes in order to experimentally
characterise the efficiency of the inlet system. We find that the coarse mode aerosol is sub-
isokinetically enhanced, with a peak enhancement at around 10 μm up to a factor of three under
typical operating conditions. Calculations show that the efficiency of collection then decreases rapidly
at larger sizes.  In order to minimise the isokinetic enhancement of coarse mode aerosol we
recommend sampling with total flow rates above 50 L min$^{-1}$; operating the inlet with the bypass fully
open helps achieve this by increasing the flow rate through the inlet nozzle. With the inlet
characterised, we also present single particle chemical information obtained from X-ray spectroscopy
analysis which allows us to group the particles into composition categories. Our intention is to use the
composition information in parallel with filter based ice nucleating particle measurements in order to
correlate composition and ice nucleating particle concentrations.





## 1 Introduction

Atmospheric aerosol particles are known to have an important effect on climate through directly scattering or absorbing solar and terrestrial radiation as well as through indirect effects such as acting as Cloud Condensation Nuclei (CCN) or Ice-Nucleating Particles (INPs) (Albrecht, 1989; DeMott et al., 2010; Haywood and Boucher, 2000; Hoose and Mohler, 2012; Lohmann and Diehl, 2006; Lohmann and Gasparini, 2017). Aerosol particles across the fine (diameter < 2 μm) and coarse (>2 μm) modes are important for these atmospheric processes.  For example, aerosol in the accumulation mode are important CCN (Seinfeld and Pandis, 2006), whereas supermicron particles are thought to contribute substantially to the INP population (Mason et al., 2016; Pruppacher and Klett, 1997)  Hence, being able to sample across the fine and coarse modes is required to understand the role aerosol play in our atmosphere. However, sampling aerosol particles without biases can be challenging, this being especially so on a fast moving aircraft (Baumgardner et al., 2011; Baumgardner and Huebert, 1993; McMurry, 2000; Wendisch and Brenguier, 2013).

It is necessary to sample aerosol from aircraft because in many cases aircraft offer the only opportunity to study aerosol and aerosol-cloud interactions at cloud relevant altitudes (Wendisch and Brenguier, 2013). However, the relatively high speeds involved present a set of unique challenges for sampling aerosol particles. This is especially so for coarse mode aerosol which are prone to both losses as well as enhancements because their high inertia inhibits their ability to follow the air stream lines when they are distorted by the aircraft fuselage and the inlet (Brockmann, 2011; McMurry, 2000; von der Weiden et al., 2009). Therefore, inlet design and characterisation becomes extremely important when sampling aerosol particles.

In this study we characterise the inlet system used for collecting filter samples (known as the Filters system) on board the UK's BAe 146 Facility for Airborne Atmospheric Measurements (FAAM) research aircraft.  This system has been used for many years, but its characterisation has been limited. Our goal in this characterisation work was to define recommendations for the use of the inlet system to minimise sampling biases and define the size limitation and the biases that exist. While the filter samples could be used for a variety of offline analyses, we have done this characterisation with two specific goals in mind: firstly, we want to use this inlet system for quantification of INP (the technique for this analysis has been described previously (Price et al., 2018) and will not be further discussed here); secondly, we have adapted and developed a technique for quantification of size distributions and size resolved composition using Scanning Electron Microscopy (SEM). We use the SEM derived size distributions in comparison with the size distributions obtained from the underwing probes to experimentally test the inlet efficiency. These experiments are underpinned by calculations which elucidate how the biases are impacted by variables such as flow speed, angle of attack and use of the bypass system. Finally we present an example of the use of the inlet for determining the size resolved composition of an aerosol sample collected from the FAAM aircraft.

## 2. Description and theoretical sampling characteristics of the filter inlet system on the Facility for Airborne Atmospheric Measurements (FAAM) aircraft

Ideally, aerosol particles would be sampled through inlets without enhancement or losses. However, this is typically not the case when sampling from aircraft, hence it is important to know how the size distribution of the aerosol particles is affected by the sampling. Generally, an aircraft moves at high velocities with respect to the air mass that it is being sampled. During sampling on the FAAM aircraft the indicated airspeed is 100 m s$^{-1}$, which yields to a true airspeed that fluctuates between 100 and 120 m s$^{-1}$. The air mass has to decelerate when passing through the inlet (Baumgardner and Huebert, 1993) and this tends to result in inertial enhancement of coarse mode aerosol. There are also losses





through the inlet system, for example, through inertial impaction at bends or gravitational settling in
horizontal sections of pipework. These inlet characteristics need to be considered if the subsequent
analysis of the aerosol samples is to be quantitative. In this section we first describe the existing inlet
system and then present theoretical calculations for the size dependent losses and enhancements.
**2.1 Description of the Filters system**
The UK's FAAM BAe-146 research aircraft has two identical inlets for sampling aerosol onto filters for
offline analysis. This inlet system was used to sample aerosol particles on board of the C-130 aircraft
before being installed on the FAAM Bae-146 (Andreae et al., 1988; Andreae et al., 2000; Talbot et al.,
1990), and it has been used to sample aerosol particles on the FAAM Bae-146 e.g. (Chou et al., 2008;
Hand et al., 2010; Price et al., 2018; Young et al., 2016). A diagram of the inlet system can be seen in
Fig. 1. The two parallel inlet and filter holder systems, which each have a nozzle whose leading edge
profile follows the criteria for aircraft engine intakes at low Mach numbers (low speeds when
compared with the speed of sound; for FAAM during sampling this is ~0.3), and it is designed to avoid
the distortion of the pressure field at the end of the nozzle, flow separation and turbulence (Andreae
et al., 1988). The inlet has a bypass to remove water droplets or ice crystals through inertial separation
and also enhance the flow rate at the inlet nozzle (Talbot et al., 1990). The flow through the bypass
(bypass flow) can be regulated using a valve and it is driven passively by the pressure differential
between the ram pressure inlet and the Venturi effect on the exhaust. After turning inside the aircraft,
the airstream containing the aerosol particles continue through the filter stack after passing a valve.
The air flow through the filter (filter flow) is measured by a mass flow meter, which reports the
standard litres sampled (273.15 k, 1013.529 hPa). The signal is integrated by an electronics unit to give
the total volume of air sampled for any given time period. There is also a valve between the pump and
the flow meter. The valve allows the inlet and pump to be isolated from the filter holder when
changing the filter. The system uses a double-flow side channel vacuum pump model SAH55 made by
Rietschie, aided by the ram effect of the aircraft. The flow rate at the inlet nozzle (total flow) is the
sum of the bypass flow and the filter flow. The inlet nozzle is located at 19.5 cm of the aircraft fuselage,
so the sampling is carried out in the free stream, outside the boundary layer.

**2.2 Sampling efficiency**
We present theoretical estimates of the losses and enhancements due to aspiration, inlet inertial
deposition, turbulent inertial deposition, inertial deposition in bends and gravitational effects in Fig
2a. We used the term 'efficiency' to define the ratio between the number concentrations of particles
after they were perturbed relative to the unperturbed value. If the efficiency is above one, the number
of particles is enhanced whereas if it is below 1, particles are lost.
The sampling efficiency of any inlet depends strongly on the flow rates and the flow regime (laminar
vs turbulent). Filter flow rates for 0.4 μm polycarbonate filters normally vary between 10 and 50 L min⁻
$^{1}$ depending on altitude (see section 2.3 for a discussion of flow rates). The bypass flow rate (when it
is fully open) can go up to 35 L min$^{-1}$ at 30 m and 22 L min$^{-1}$ at 6 km, but it is not measured routinely.
In the 2.5 cm diameter section of the inlet, the Reynolds number (Re) is below the turbulent regime
threshold (Re > 4000) for flow rates below 65 L min$^{-1}$. For larger values of Re, the flow starts becoming
turbulent. At the inlet nozzle, where the diameter 0.7 cm, Re is above 4000 for flow rates above 20 L
min$^{-1}$, so the flow is briefly in the turbulent regime at the inlet for most sampling conditions. Fully
characterising the losses and enhancements of aerosol particles passing through the inlet is very





challenging since there are several aerosol size dependent mechanisms than can enhance or diminish
the amount of aerosol particles that arrive at the filter.
Here we have considered the most important of these mechanisms in order to estimate the inlet
efficiency (see Fig. 2a) for a total flow rate of 50 L min$^{-1}$. These loss mechanisms and their importance
in this inlet system are defined as follows (a discussion on the choice of equations and how they have
been applied can be found in Appendix A):
*Aspiration efficiency* accounts for the fact that the speed of the sampled air mass ($U_0$) and the speed
of the air through the beginning of the inlet nozzle (U) are different. In the case of the filter inlet
system on the FAAM aircraft, the speed of the air mass is greater than the speed through the inlet
(sub-isokinetic conditions), which leads to an enhancement of larger aerosol particles. Here, we have
used the empirical equation as develop in Belyaev and Levin (1972) and Belyaev and Levin (1974). As
one can see in Fig. 2a this mechanism enhances aerosol particles, tending to 1 for small diameters and
to the ratio $U/U_0$ for large ones.
*Inlet inertial deposition* is the inertial loss of aerosol particles within the nozzle because the flow
expands inside the nozzle and the trajectory is therefore bent towards the wall. It has been
characterised using the equation given in Liu et al. (1989) which quantifies this effect. In Fig. 2a one
can see that it produces some losses, with a minimum efficiency of down to 50% for sizes about 6 µm,
without affecting the lower and upper limit of the aerosol size.
*Turbulent inertial deposition* happens when some particles are collected by the wall due to
turbulences in the system. In our case, this occurs throughout the whole inlet system for flow rates
above 65 L min$^{-1}$ and only occurs in the inlet nozzle for flow rates below this threshold. We have used
the equation given by Brockmann (2011) in order to account for this mechanism, which can be seen
in Fig. 2a. This mechanism gradually decreases the efficiency for aerosol particles above 5 µm.
*Bending inertial deposition* of aerosol particles is important in this inlet system at the 45$^{o}$ bend, where
some particles are not able to follow air streamlines at bend. We have characterised these losses using
the equation given in (Brockmann, 2011). This efficiency mechanism, which can be seen in Fig. 2a,
adds a size cut off with a D50 value at ~25 µm.
*Gravitational settling* of aerosol particles was considered using the equations developed in Heyder
and Gebhart (1977) and Thomas (1958), as stated in Brockmann (2011). This efficiency mechanism
adds another size cut off with a D50 value of 35 µm, as one can see in Fig. 2a.
*Diffusional efficiency and filter collection efficiency* have not been included in Fig. 2. The first
mechanism has been calculated using the analytical equation given by Gormley and Kennedy (1948),
but it is not shown since it is very close to 1 for all the considered size range. For the filter types and
pore sizes we used, filter collection efficiency is also close to a 100% across the relevant size range
(Lindsley, 2016; Soo et al., 2016).
*Anisoaxial losses* are losses produced by the fact that the inlet is not aligned with the velocity of the
air mass, being offset by an angle, θ (related to the angle of attack). The anisoaxial sampling can affect
the sub-isokinetic efficiency, but using the equations given by Hangal and Willeke (1990a), we
calculated that this effect is minimal for our conditions. In addition, anisoaxial sampling can lead to
inertial losses when particles impact the inner walls of the inlet. This phenomena has been quantified
using the equations in Hangal and Willeke (1990b) and the results can be seen in Fig. 3. As one can
see, this efficiency mechanism adds an additional cut off for large aerosol particles (with values of D50
down to ~20 µm), depending on the value of the sampling angle.



One can see all the efficiency mechanisms combined for four different flow rates in Fig. 2b. These have
been derived by multiplying all the efficiencies for the individual mechanisms. This overall efficiency
is the ratio between the particles that reach the filter and the particles in the ambient air.  The
sampling efficiency for the submicron aerosol is close to 1. At sizes above 1 μm, the different loss
mechanisms become increasingly significant. For the range of flow rates considered, the efficiency
approaches zero between 20 and 50 μm, with D50 values in between ~13 and ~33 μm (although these
values could be lower under certain values of angles of attack if considering the anisoaxial losses of
from Fig. 3, which haven't been included). For the 80 L min$^{-1}$ case, the flow is turbulent through all the
pipe, leading to enhanced losses of coarse aerosol particles which partially compensate the sub-
isokinetic enhancement of the system.
One can also see that the sub-isokinetic enhancement of large aerosol particles increases when
decreasing the flow rate of the system. This effect is about a factor 3.5 for 10 μm particles when
sampling at 15 L min$^{-1}$, but only a factor of two at 50 L min$^{-1}$. The sub-isokinetic enhancement can be
mitigated using the bypass, which enhances the flow through the nozzle. This can be seen in Fig. 2c
where one can see a comparison between the total efficiency of a 20 L min$^{-1}$ flow rate through the
filters with no bypass flow and the same case when the bypass is open. Since the considered bypass
flow is comparable to the flow rate through the filters, the difference between the total flows for the
two cases is approximately a factor 2. As a consequence, the maximum sub-isokinetic enhancement
of large aerosol particles is almost a factor 2 larger when sampling with the bypass closed. Hence, the
sub-isokinetic enhancement can be reduced by keeping the bypass fully open.

**2.3 Sampling flow rate**
Here we show flow rate data from four field campaigns in order to examine how the flow rate of the
filter inlet system varied based on different factors. We have used the data collected during the ICE-D
campaign, in Cape Verde during August 2015 (Price et al., 2018). The rest of the data is from some
flight test carried out during 2017 and 2018, and three field campaigns. The first one was EMERGE,
based in south east England, in July 2017. The second one was VANAHAEIM, based in Iceland in
October 2017. The last campaign was MACSSIMIZE, based in Alaska in 2018. The flow rate of the inlet
system is known to vary with altitude, with a lower flow rate at high altitudes because of the reduced
pressure differential across the filter and the fact that the pump efficiency decreases at low pressure.
In addition, it changes depending on the filter type and the pore size.
In Fig. 4, where all the flow rate data has been presented, one can see that the flow rate tends to
decrease with altitude and change with filter type as expected, but the flow rates are not consistent
for each altitude and filter type, varying up to a factor two for each filter type/line/altitude/campaign.
The filter type effect on flow rate can be seen in Fig. 4, where the average flow rate for 0.4 μm
polycarbonate filters is about twice the flow rate of the 0.45 μm PTFE filters.  In order to investigate
the inconsistency in the flow rate at each altitude, we analysed the flow rate data by comparing it with
different parameters (ambient air and cabin temperature, ambient air and cabin pressure, wind
direction and speed with respect to the aircraft movement), but there was no correlation with any of
these variables. Different mesh supports were used, but this does not affect the flow rate significantly
according to some ground based tests. We checked the flow rate through each sampling period and
found it did not change over time on a particular filter set (even after stopping the sampling and
starting it again). In addition, we performed some tests on the ground and during flights to study the
effect of potential leaks by inserting paper disks of the same dimension as the filters in the filter
holders and found no evidence of leaks in the system.



We conclude that this variability in the flow rate comes from variability in the pump performance in
combination with subtle differences in individual filter pairs. The side displacement pump is not the
ideal pump for this system and operates at its maximum capacity.  Hence, we suggest that to improve
the performance of the system that flow rates are actively controlled and also the side displacement
pump is replaced with a more appropriate design. This would also have the advantage that flow rates
would be maintained at smaller pressure drops and allow sampling at higher altitudes.

**3. FAAM underwing optical particle counters**
Later in the paper we compare results from the underwing optical particle counters with our electron
microspore derived size distributions, hence we describe the optical instruments here. The BAe-146
FAAM research aircraft operates underwing optical particle counters to measure aerosol size
distributions. These include the Passive Cavity Aerosol Spectrometer Probe 100-X (PCASP) and the
Cloud Droplet Probe (CDP). The PCASP measures particles with diameters in the approximate range
0.1-3 µm and the CDP measures the particles with diameters in the range of 2-50 µm. These
instruments are placed outside the aircraft fuselage, below the wings. These instruments and the
methods for calibration are described in (Rosenberg et al., 2012).
The instruments were calibrated and had optical property corrections applied as per Rosenberg et al.
(2012). We used a refractive index of 1.56 + 0i and a spherical approximation (Mie theory) in the
optical property corrections. In Fig. 5, one can see a sensitivity test on the refractive index value we
used in order to examine how variability in refractive index affect the bin centres position, their width,
and therefore the size distribution obtained from the PCASP and CDP. As one can see in Fig. 5a,
modification of the real part of the refractive index from 1.5 to 1.7 can change the position of the
PCASP bin centres up to a factor 1.5, but its effect on the CDP is not significant. When varying the
imaginary part of the refractive index from 0 to 0.01, the bin centre positions of the first half of the
range of the PCASP and CDP do not change but it can change the position of the bins of the end of the
range of both instruments (less than a factor 1.5). However, for the purposes of this work, the
differences produced by the variation in the refractive index are not large enough to modify the
conclusions of the analysis, therefore we use a value of 1.56 + 0i.
The chosen refractive index range for this sensitivity analysis can be justified on the basis that the SEM
compositional analysis showed that the composition of the aerosol samples used in this study was
very heterogeneous, dominated by carbonaceous particles (biogenic, organic and black carbon) and
with some contributions of mineral dust and other particle types. Values of the real part of the
refractive index in the 1.5 to 1.6 range are compatible with sodium chloride and ammonium sulphate
(Seinfeld and Pandis, 2006), as well as most mineral dusts (McConnell et al., 2010). The range is very
close to values for the real part of the refractive index of organic carbon but below the values for black
carbon (Kim et al., 2015). As a consequence, the refractive index choice might not be accurate for a
black carbon dominated sample. However, black carbon is not likely to dominate in the size range
where a value of the real part of the refractive index of 1.7 dramatically changes the size distribution
(diameters above 0.5 µm) (Seinfeld and Pandis, 2006), so our refractive index choice is valid.  In Fig.
5b one can see that changing the imaginary part of the refractive index from 0 to 0.01 only produces
small changes in the distribution. The imaginary part of the refractive index of many aerosol types as
sodium chloride, sulphates and mineral dust falls within the shown range (Seinfeld and Pandis, 2006),
(McConnell et al., 2010). For values of the imaginary part of the refractive index above 0.01 (not shown
in the image), the size distribution dramatically changes for sizes above 1 µm (but not for smaller
values of it), overlapping and disagreeing with the CDP. However, values above 0.01 in the imaginary



part of the refractive index can only happen in black carbon, which will dominate only in the submicron
sizes (Seinfeld and Pandis, 2006). The submicron part of the size distribution doesn't change for values
of the imaginary part of the refractive index above 0.01, so our refractive index choice is still
acceptable even for samples with significant contributions from black carbon in submicron sizes.
For the PCASP-CDP, we have considered two uncertainty sources. The first one is the Poisson counting
uncertainty in the number of particles in each bin and the second one is the uncertainty in the bin
width that is given by the applied optical property corrections. Both sources have been propagated in
order to obtain the errors of d$N$/dlog$D$p and d$A$/dlog$D$p. Other uncertainties such as the refractive
index assumption or particle shape effect, as well as the uncertainty in the bin position haven't been
shown in this study. Sampling biases haven't been quantified or corrected yet so they haven't been
included. The size distributions produced by the PCASP-CDP have been taken as a reference value for
the purposes of this study.

**4. Scanning Electron Microscopy technique for aerosol characterization**
Scanning Electron Microscopy is used in order to study composition and morphology of aerosol
particles, in a similar way to previous works such as Chou et al. (2008), Young et al. (2016) and Price
et al. (2018). We use a Tescan VEGA3 XM scanning electron microscope (SEM) fitted with an X-max
150 SDD Energy-Dispersive X-ray Spectroscopy (EDS) system controlled by an Aztec 3.3 software by
Oxford Instruments, at the Leeds Electron Microscopy and Spectroscopy Centre (LEMAS) at the
University of Leeds. In order to get data from thousands of particles in an efficient way, data collection
was controlled by the AztecFeature software expansion.
Aerosol particles were collected with the filter inlet of the FAAM aircraft on polycarbonate track
etched filters with 0.4 μm pores (Whatman, Nucleopore). Samples for SEM are usually coated with
conductive materials in order to prevent the accumulation of charging on the sample surface (Egerton,
2005). For aerosol studies, materials like gold (Hand et al., 2010), platinum (Chou et al., 2008), or
evaporated carbon (Krejci et al., 2005; Reid et al., 2003; Young et al., 2016) have been used. When it
comes to choosing which signal to detect, some previous studies used backscattered electrons (Gao
et al., 2007; Price et al., 2018; Reid et al., 2003; Young et al., 2016) and some others choose secondary
electrons (Kandler et al., 2007; Kandler et al., 2011; Krejci et al., 2005). We started the development
of this analysis using a carbon coating and the backscattered electron detector. This technique
produced reproducible images and almost no artefacts from the pore edges, consistent with Gao et
al. (2007). However, we noticed that we were undercounting a significant fraction of the small carbon
based particles, which looked transparent under the backscattered electron imaging but not under
the secondary electron detector, as seen in Fig. 6. This likely happened because the contrast in the
secondary electron images mainly depends on the topography of the sample whereas the contrast in
the backscattered electron images depends on the mean atomic number of each sample phase
(Egerton, 2005). Since the polycarbonate filters are made of C and O, particles containing only these
elements in a similar proportion to the background did not exhibit a high contrast under the
backscattered electron detector (Laskin and Cowin, 2001). However, when using secondary electron
imaging with carbon coatings, images were less reproducible and contained artefacts from the pore
edges, probably resulting from charging or topographical effects. We found that coating the samples
with 30 nm of iridium helps to improve the secondary electron image reproducibility and reduced the
pore edge artefacts as well as allowing us to locate small organic particles. An increase in the size of
the particle as a consequence of the coating may introduce an uncertainty in the size of the smallest



particles. An additional advantage of using Ir is that the energy dispersive X-ray spectrum of Ir does
not overlap greatly with the elements of interest.
In the SEM the sample was positioned at a working distance of 15 mm. The SEM's electron beam had
an accelerating voltage of 20 KeV and a spot size chosen to produce the optimum number of input
counts in the EDS detector. Images are taken at two different magnifications with a pixel dwell time
of 10 μs and a resolution of 1024 x 960 pixels per image. High magnification images (x 5000 or similar)
were used to identify particles down to 0.3 or 0.2 μm depending on the sample, and medium
magnification images (x1500 or similar) are used to identify particles down to 1 μm. A brightness
threshold with upper and lower limits that correspond to pixels of certain shades of grey was manually
adjusted for each image by the operator to discriminate particles from the background. Based on the
manually set brightness threshold, AztecFeature identified the pixels that fall within the limits as
aerosol particles and calculated several morphological properties of the particle as cross sectional
area, length, perimeter, aspect ratio, shape factor or equivalent circular diameter. The equivalent
circular diameter is defined as $\sqrt{(4\,A\,\pi^{-1})}$, where $A$ the cross sectional area of the aerosol particle.
For this analysis we placed a section of the 47 mm filter on a 25 mm stub. In order to collect
morphological and chemical information from a few thousand particles, we only scanned a fraction of
the filter. We collected information from 5 to 20 different areas, and each area consisted of a montage
of several SEM images. The areas were chosen by the user from all over the surface of the selected
fraction of the filter, since aerosol particles were evenly distributed all over the central ~30mm of the
filter (the area which exposed to the air) as one can see in Fig. 7. Each area was selected in the
software, manually adjusting the particle detection threshold. The Z position of the stage was also
adjusted manually for each image in order to produce properly focused images. After doing this, the
image scanning and EDS acquisition was performed in an automated way. Morphological information
was recorded for all particles with an ECD greater than the specified size threshold (typically 0.2 or 0.3
μm).
EDS analysis was restricted to the first 12 or 15 particles detected in each image. This reduces the
likelihood of charging problems caused by exposure to the electron beam. The software performed
EDS in the centre of the particles, obtaining around 50,000 counts per particle. The raw data for any
given particle were matrix corrected and normalised by the AZtec software to produce element weight
percent values with a sum total of 100%, using a value of the confidence interval of 2 (a further
discussion on the confidence interval can be seen in Fig. S1). Then particles were categorised based
on their chemical composition using a classification scheme which can be created and modified within
the AztecFeature software. The characteristic X-rays taken at one point are emitted by a certain
interaction volume which is bigger than some of the analysed particles (typically < 2μm³, decreasing
with atomic number and increasing with incident electron energy). As a consequence, a part of the X-
ray counts attributed to each particle come from the background (C and O from the polycarbonate
filter and Ir from the coating) and the weight percentages obtained from the X-ray spectra do not
match the actual weight percentages of the particle itself. As a consequence, when categorising the
particles based on their composition, we only use the presence or absence of certain elements, and
the ratio between the weight percentages of non-background elements. The classification scheme
works by checking if the composition of each particle falls within a range of values which are manually
defined by the user. Particles not matching the first set of rules are tested again for a second set of
rules, and so on, until reaching the last set of rules. A few sets of rules can be merged into a category.
In the supplementary information (Fig. S3), we give the details of the 32 sets of rules used, which are
then summarised into 10 composition categories. A description of the most abundant elements in
each category and an interpretation of these categories is included in Sect. 5.





The detection of particles has certain limitations. The edges of the pores can look brighter than the
rest of the filter in the SE images (probably because they consist of a larger surface area from which
secondary electrons can be generated, hence a larger signal). As a consequence, they can look like
~0.2 μm particles, which is the main reason why particles below 0.3-0.2 μm (depending on the sample)
are not included in this analysis. These artefacts had a chemical composition similar to the filter, so
they were labelled as "Carbonaceous" by the classification scheme, falling at the same category as
most biogenic and black carbon particles. However, these artefacts were only around 1 to 10 percent
depending on the sample. If they appear in larger quantities, they can be removed manually after or
during the analysis. Another limitation arises from the fact that some aerosol particles did not have
sufficient brightness in the SE image and were not detected as a particle. This happens more
frequently for smaller particles, but it can also happen with some larger particles, particularly if they
are only composed of Na and Cl or S. This issue can be addressed if necessary by setting a very low
limit of detection, which adds lots of artefacts as well as the low brightness particles, and then
removing the artefacts manually (the artefacts can be easily identified by the user). In other infrequent
instances, only a fraction of the particle had a brightness above the threshold, so they were detected
as a smaller particle or multiple smaller particles, or if two particles are close enough, they can be
detected as a single larger particle. However, we feel that in the vast majority of the cases a
representative cross sectional area of the particle was picked by the software.
Blank polycarbonate filters can contain some particles on them from manufacturing or transport
before being exposed to the air. In addition, handling and preparing the filters can introduce additional
particles to it. In order to assess these artefacts, we scanned a few clean blank filters. We also
examined a filter that had been brought to the flight, loaded in the inlet system (but not exposed to a
flow of air), and then stored at -18 °C for a few months (like most of the aerosol samples on filters).
The results of both the handling blank and the blank can be seen in Fig. S2. The number of particles is
very low, typically about the order of magnitude of one particle per 100 by 100 μm square, which is
well below the typical particle loading on a filter exposed to the atmosphere. In Fig. S2 one can see
that the vast majority of particles found in both blank filters and the handling blank belong to the
metallic rich category. However, further examination of the composition of these particles revealed
that almost all of them were Cr rich particles (about 97 % in the case of the blank filters and about
96% in the case of the handling blank). As a consequence, we excluded all the Cr rich particles from
the analysis of atmospheric aerosol (it was only ever a very minor component). By doing this, we make
sure that we excluded more than half of the artefacts of the analysis. The composition of the particles
present in the blank filters and in the handling blank was very similar, suggesting that most of these
artefacts are not produced by the loading, manipulation and storage of the filter. However, there was
a very small but significant contribution of mineral dust origin particles (Al-Si rich, SI rich and Si only)
for sizes in between 0.7 and 5 μm in the handling blank, which should be taken into consideration.

**5. Inlet characterisation and sampling efficiency using Scanning Electron Microscopy**
In order to experimentally test the inlet efficiency, to complement the efficiency calculations
presented in Section 2.2, we have used SEM to quantify the size distribution of particles collected on
filters (Sect. 4) and compare this with the measurements from the under-wing optical probes (Sect.
3). The calculations in Sect. 2.2 suggest that there is an enhancement of the coarse mode aerosol
particles, which is larger when sampling with the bypass closed. To test this we have collected aerosol
onto 0.4 μm pore size polycarbonate filter in both lines in parallel. In one of the lines, the bypass was
kept open, and in other line the bypass was kept closed. Using our SEM approach described in the
Sect. 4, we calculated the size distribution of the aerosol particles on top of each filter. We compared





these size distributions with the ones measured by the underwing optical probes (PCASP-CDP), as
described in Sect. 3. We performed the test twice in two different test flights based in the UK.
The results of these comparisons can be seen in Fig. 8 and Fig. 9 for both number and surface area size
distribution. There are some discrepancies between the optical probes and the SEM size distributions
from the filters, which has also been reported in previous works (Chou et al., 2008), (Price et al., 2018;
Young et al., 2016). There is significant disagreement between the submicron particles detected on
top of the filter and the submicron particles detected in situ by the PCASP-CDP in Fig. 9a in a very
similar way to Young et al. (2016). Since the 0.4 μm polycarbonate filters have a high collection
efficiency at these length scales (Lindsley, 2016; Soo et al., 2016), the disagreement at the submicron
regime could be produced by several effects. Some small particles that may not have sufficient
brightness to be detected might produce some undercounting, despite the fact we made efforts to
mitigate this problem. Also, volatilization of certain type of aerosol particles (which are more abundant
in the submicron fraction (Seinfeld and Pandis, 2006)) can happen during heating or sampling (Bergin
et al., 1997; Hyuk Kim, 2015; Nessler et al., 2003) and this effect could be enhanced by the fact that
samples are exposed to high vacuum during the SEM analysis. The presence of nitrates could be tested
using an aerosol mass spectrometer. In addition, the SEM techniques measure the dry diameter and
the optical probes measure the aerosol diameter at ambient humidity. This hygroscopic effect is
known to shift the dry size distributions to smaller sizes, which could also explain part of the
disagreement (Nessler et al., 2003; Young et al., 2016). Disagreement in the measurements can be
also produced by the fact that the techniques are measuring different diameters; optical diameter in
the case of the PCASP-CDP and circular equivalent geometric diameter in the case of the SEM.
One can see that the concentration of aerosol particles measured by the SEM on the filters was higher
than the particles detected by the optical probes for sizes above ~8 μm in Fig. 8 and Fig. 9. These
results are consistent with Price et al. (2018), where they observed a similar enhancement of large
aerosol particles in two mineral dust dominated samples collected close to Cape Verde. In addition,
the enhancement was larger when sampling under closed bypass conditions. The results of these
comparisons are in agreement with the theoretical calculations in Sect. 2.2.
In Fig. 10 we have presented some other SEM size distributions compared with the PCASP-CDP data
from three different aerosol samples in contrasting locations. Since these data were taken during the
scientific field campaigns and not test flights, we only collected one polycarbonate filter for SEM since
the other line was used for INPs analysis on Teflon filters. All the sampling was done with the bypass
open. The agreement between the optical and SEM obtained size distributions in Fig. 10 is similar to
the one in Fig. 8 and Fig. 9 (for the open bypass case). One can observe in Fig. 10c that there was a
loss of particles smaller than ~0.5 μm and also that in Fig. 10a and Fig. 10b there was an enhancement
of the coarse mode.
The data shown in this section have been obtained in very different locations (South England in the
case of Fig. 8, Fig. 9 and Fig. 10a, Iceland in the case of Fig. 10b and north Alaska in the case of Fig. 10c.
As a consequence, the studied aerosol samples are very different in both morphology and chemistry.
From the comparisons, we can state that the sampling carried out by the filter system has certain
biases, but it captures particles with a size distribution with similar features to the ones measured by
the underwing optical probes.



**6. Recommendations for aerosol sampling with the Filters system on the FAAM aircraft**

Based on the calculations in Sect. 2.1, we suggest keeping the total flow rate (including the flow through the filters measured by the electronics box plus the bypass flow, which can be between 20 and 35 L min$^{-1}$) above 50 L min$^{-1}$. Below this range, the sub-isokinetic enhancement of large aerosol particles is above a factor 2, according to the calculations in Sect. 2.2 that can be seen in Fig. 2b. For total flow rates above 65 L min$^{-1}$, the flow becomes turbulent throughout the line, which associated losses. However, the calculations shown in Fig. 2c indicate that the combination of the isokinetic enhancements and turbulent losses at 80 L min$^{-1}$ lead to a reasonably representative sampling, but when it reaches 150 L min$^{-1}$, the position of the D50 drops to 6.5 µm (not shown in the graph) so such a high flow rate would not be ideal if the user wants to sample coarse aerosol particles. Hence, we recommend an operational upper limit of 80 L min$^{-1}$. For 0.45 µm PTFE filters and the 0.4 µm polycarbonate filters presented in Fig. 4, sampling close to this flow rate range is often achievable by keeping the bypass open, since this increases the total flow rate and brings it closer to the suggested range, as one can see in Fig. 2c. If other filter types are used, these results should be taken in consideration when choosing the pore size or equivalent pore size in order to avoid dramatic sampling biases.

We already mentioned in Sect. 2.3 that we recommend replacing the side displacement pump with a design that would provide a greater pressure drop. In addition, we also recommend that the bypass flow rate is also routinely measured and controlled in order that the flow at the inlet nozzle can be optimised while sampling.

**7. SEM compositional categories**

Here we describe the 10 categories we have used in our compositional analysis, which are a summary of the 32 rules described in the supplementary information. The approach has some similarities with the ones in previous studies (Chou et al., 2008; Hand et al., 2010; Kandler et al., 2011; Krejci et al., 2005; Young et al., 2016), but it is distinct. Because of the fact that the filter is made of C and O, background elements (C and O) were detected in all the particles. Particles in each category can contain smaller amounts of other elements apart from the specified ones. This classification scheme has been designed a posteriori to categorise the vast majority of the aerosol particles in the three field campaigns previously described and some ground collected samples in the UK and Barbados. The main limitation of the classification scheme is the difficulty to categorise internally mixed particles. The algorithm has been built in a way it can identify mixtures of mineral dust and sodium chloride (they appear as mineral dust but they could be split into a different category if necessary) and sulphate or nitrate ageing on sodium chloride (they appear as Na rich but it could also be split into a different category). However, other mixtures of aerosol wouldn't be identified, and they would be categorised by the main component in the internal mixture in most cases.

7.1. Carbonaceous

The particles in this category contained only background elements (C and O). The components of the carbonaceous particles consist in either black carbon from combustion processes or organic material, which can be either directly emitted from sources or produced by atmospheric reactions (Seinfeld and Pandis, 2006). Particles containing certain amount of K and P in addition to the background elements were also accepted in these category. These elements are consistent with biogenic origin aerosol particles (Artaxo and Hansson, 1995). Distinction between organic and black carbon aerosol unfortunately could not reliably be done. Since N is not analysed in our SEM set up, any nitrate aerosol particle would fall into this category if it is on the filter. However, since these particles are semi-





volatile, some of these aerosol particles would not resist the low pressure of the SEM chamber. This
could be further investigated in the future.
7.2. S rich
Aerosol particles in this category contained a substantial amount of S. These EDS signals are
compatible with sulphate aerosol particles, which are solid or liquid sulphuric acid particles (Kumar
and Francisco, 2017). In the same way as the nitrates, this particles are semi-volatile and some of them
might not resist the low pressure of the SEM chamber.
7.3 Metal rich
The composition of particles in this category is dominated by one of the following metals: Fe, Cu, Pb,
Al, Ti, Zn or Mn. These EDS signatures are compatible with metallic oxides or other metal rich particles.
These metal containing particles can originate from both natural sources and anthropogenic sources.
Some metallic oxides are common crustal materials that could go into the atmosphere but are also
produced during some combustion processes (Seinfeld and Pandis, 2006). In addition, many types of
metal and metallic derivatives particles are produced as component of industrial emissions and other
anthropogenic activities (Buckle et al., 1986), (Fomba et al., 2015).
7.4. Na rich
Sodium chloride particles are the main component of the sea spray aerosol particles which are emitted
through wave breaking processes (Cochran et al., 2017). These particles can age in the atmosphere by
reacting with atmospheric components such as sulphuric or nitric acid (Graedel and Keene, 1995),
(Seinfeld and Pandis, 2006). As a consequence of this reaction, a part of their Cl content will end up in
the gaseous phase (as HCl), leading to an apparent chlorine deficit in the aged sea spray aerosol
particles. Particles in this category have an EDS signature compatible with sea spray aerosol particles
since they are identified by the presence of Na, containing in most cases Cl and/or S (N is not included
in our SEM analysis).
7.5 Cl rich
Particles in this category contained mainly Cl and sometimes also K but never Na, so they are not
sodium chlorine particles. Significant concentrations of Cl and metals in aerosol particles have been
linked to industrial activities and automobile emissions (Paciga et al., 1975), whereas Cl and K in
aerosol particles could be originated by the use of fertilisers (Angyal et al., 2010), or emitted during
pyrotechnic events (Crespo et al., 2012).
7.6 Ca rich
The composition of the particles in this category is dominated by Ca. In this category, particles
containing only Ca (plus C and O, the background elements) are consistent with calcium carbonate, a
major component of mineral dust (Gibson et al., 2006). If other elements such as Mg and S are present,
the signature of the particles compatible with some mineral origin elements as gypsum and dolomite
respectively. In addition, presence of minor amounts of Si, Al and other elements could indicate mixing
of these Ca rich particles with some other soil components as silicates. However, since Ca is a biogenic
element, we cannot discard the biogenic origin of some of the Ca-rich particles (Krejci et al., 2005).
7.7 Al-Si rich
Particles in the Al-Si rich category were detected by the presence of Al and Si as major elements. Very
often, this particles also contained smaller amounts of Na, Mg, K, Ca, Ti, Mn and Fe. Particles in this



category are very likely to have mineral origin and are commonly described as aluminosilicates which
include a range of silicates such as feldspars and clays (Chou et al., 2008; Hand et al., 2010). Mixed
mineral origin particles containing both Al and Si can also fall into this category. Strong presence of Na
and Cl could indicate internal mixing with some sea spray aerosol, whereas a strong S presence could
indicate atmospheric acid ageing.
**7.8 Si only**
The particles in this category contained only Si apart from the background elements. Particles in this
category are very likely to be a silica polymorph (mainly quartz), one of the major components of the
earth's crust. Since we cannot determine if the C signal in the EDS of these particles is produced from
the background or from the particle itself, a particle containing only C, Si and O would fall into this
category, however, mineral phases containing these elements are extremely rare.
**7.9 Si rich**
The composition of these particles was dominated by Si, and other elements Na, Mg, K, Ca, Ti, Mn and
Fe. The main difference with the particles in Sect. 7.7 is that the ones described here didn't contain Al
above the limit of detection. The EDS signal of particles in this category is compatible with any silicate
that does not contain Al as a major component in its phase such as talc or olivine. The only exception
is quartz, which falls in the 'Si only' category described above. They could also be internal mixtures of
silica or silicates without aluminium as a major component in its phase. Because of the high limit of
detection of the Al (See the SI), some particles in this category could contain small amounts of Al, and
should belong to Al-Si rich category. As in the Al-Si rich particles case, strong presence of Na and Cl
could indicate internal mixing with some sea spray aerosol, whereas a strong S presence could indicate
atmospheric acid ageing.
Some of these categories could be further grouped. For example, the particles in the Ca rich, Al-Si
rich, Si only and Si rich categories could be considered as "mineral dust". However, if the sample
contains ash from combustion processes or volcanic origin, it will also appear in these last categories
since its composition is similar to mineral dust (Chen et al., 2012; Nakagawa and Ohba, 2003).
**8. Application to a sample collected from the atmosphere above S.E. England**
The SEM technique described in Sect. 4 has been applied to samples collected from the FAAM aircraft
in various locations. Here we show an example of some of the capabilities of the developed technique
applied to one of the samples collected in S. E. England; the resulting size resolved composition is
shown in Fig. 11. The fraction of particles corresponding to each compositional category described in
Sect. 7 for each size can be seen in Fig. 11a. The SEM size distribution of each composition category
can be seen in Fig. 11b. By looking at this analysis, one can see that the sample was clearly dominated
by carbonaceous aerosol particles in all the sizes, but there was a clear mineral dust mode (Si only, Si
rich Al-Si rich and Ca rich) and some smaller contributions of other aerosol types (metal rich and S
rich). A potentially useful application of the size resolved composition is calculating the surface area
or mass of an individual component of a heterogeneous aerosol. As an example, we have grouped the
mineral dust categories Si only, Si rich Al-Si rich and Ca rich to produce the surface area size
distribution of mineral dust (and potentially ash) in Fig. 11c.
There are very few ways to obtain the size-resolved composition of an aerosol sample. Single particle
laser based mass spectroscopy has been used in order to obtain the size-resolved composition of
aerosol samples, both on the ground and in an aircraft (Pratt and Prather, 2012), (Wendisch and
Brenguier, 2013). Examples of this are instruments like the Particle Analysis by Laser Mass



Spectrometer (PALMS) (Cziczo et al., 2006; Thomson et al., 2000), the Aerosol Time-Of-Flight Mass
Spectrometer (ATOFMS) (Pratt et al., 2009), or the Aircraft-Based Laser Ablation Aerosol Mass
Spectrometer (ALABAMA) (Brands et al., 2011). However, a limitation of these techniques is that they
focus on the fine mode, with limited information about the coarse mode. A size-resolved
compositional analysis as the one described here is able to obtain the accumulation and coarse mode
size distribution of mineral dust in most aerosol samples, even in the ones dominated by other species,
as shown in Fig. 11. In addition, this approach can be used to get size-dependent information about
the shape factor and other morphological properties of the mineral dust, as well as ratios in between
the element concentrations.
**9. Conclusions**
In this work we have characterised the filter inlet system on board the BAe-146 UK FAAM research
aircraft which is used for the collection of atmospheric aerosol particles for off line analysis. Our
primary goal is to use this inlet system for quantification of INP concentrations and size resolved
composition measurements, but it could also be used to derive other quantities with other analytical
techniques.
In order to characterise the inlet system we made use of an electron microscope technique to study
the inlet efficiency, by comparing the SEM size distributions with the in situ size distributions
measured with underwing optical probes (PCASP-CDP). The sub-isokinetic enhancement of large
aerosol particles predicted by the calculations in Sect. 2.2 was observed in these comparisons. We also
experimentally verify that this enhancement is minimised by operating the inlet with the bypass open
which maximised the flow rate through the inlet nozzle. In addition, we note that we performed tests
with three very different aerosol distributions and the size distribution of the particles on the filters
was comparable to those measured by the underwing optical probes. Overall, the inlet tends to
enhance the concentration of aerosol in the coarse mode with a peak enhancement at ~10 μm. The
inlet efficiency decreases rapidly for sizes above about 20 μm and becomes highly dependent upon
the specifics of the sampling such as flow rates and angle of attack. Based on these tests we
recommend that the total flow rates at the nozzle are maintained at between 50 and 80 l min$^{-1}$, and
also that improvements are made to the pump and bypass flow control (see Sect. 2.3).
We also established an SEM technique to determine the size resolved composition of the aerosol
sample. Each particle can be categorized based on its chemical composition using a custom made
classification scheme. Using this technique we showed that the filter system on board of the BAe-146
spreads the particles evenly across the filter surface, which is necessary for the SEM size distribution
analysis.
Having a well characterised inlet allows us to sample aerosol particles up to around 20 μm with
knowledge of the likely biases from the aircraft. Hence, we can use this inlet system to collect aerosol
for offline analysis at altitudes which are relevant for clouds. For example, this may allow us to use the
size resolved aerosol composition to quantify the size distribution of individual aerosol components
at a particular location and combine this information with INP measurements to quantify the surface
area normalised ice nucleating ability of a specific class of aerosol.





**Appendix A: discussion of the inlet efficiency calculations**

Here we include a further description of the efficiency mechanisms used in the inlet model described in Fig. 2 and discuss the choice of the equations and their limits of validity:

*Aspiration efficiency* accounts for the fact that the speed of the sampled air mass ($U_0$), and the speed of the air through the beginning of the nozzle (U) are different. When these two speeds are equal, the sampling is called "isokinetic", whereas when the speeds don't match, the sampling is called super isokinetic or sub-isokinetic depending on if $U_0$ is smaller or larger than U respectively. In our case, the air mass moves at the flying speed, which varies with the altitude (110 m s$^{-1}$ is a typical value for sampling altitudes), and the speed at the start of the inlet is almost always below 35 m s$^{-1}$ (sub-isokinetic conditions). As a consequence, some air streamlines will be forced around the inlet, while high inertia particles won't, which will lead to an aspiration efficiency above 1 for coarse mode aerosol particles. This enhancement is greater for large particles due to their large inertia which makes difficult their ability to follow the air streamlines. The enhancement reaches a maximum value of $U_0/U$ in its high diameter limit (when none of the particles in the sampled air mass follow the streamlines that escape from the inlet and all of them are sampled). The aspiration efficiency tends to 1 (no enhancement) for small diameters.

This behaviour has been characterised by several studies (we will only look at the sub-isokinetic range of the equations since it is impossible to reach the super isokinetic range during flight). An empirical equation was developed based on laboratory experiment by Belyaev and Levin (1972) and Belyaev and Levin (1974) (referred as B&L) for certain range of $U/U_0$ ratio and Stokes number. However, for ratios below its experimental range ($U/U_0 > 0.2$), the B&L function doesn't make physical sense since it converges to values above 1 for small particle sizes. The aircraft inlet system works at smaller $U/U_0$ ratios sometimes, so this function is not very accurate to describe the behaviour of the system in such conditions. Liu et al. (1989) developed another function (referred as LZK) by means of a numerical simulation based on computational fluid mechanics. The $U/U_0$ ratio and Stokes number valid range is wider than the B&L expression (down to 0.1). It agrees with the B&L expression in the $U/U_0$ ratio the latter was developed for. For smaller values of the ratio, the LZK function are believed to be more accurate, since it predicts the known physical behaviour (no sub-isokinetic enhancement for small particle sizes). It reaches $U/U_0$ ratios down to 0.2, which is enough to cover most of the total flow rates achieved in the inlet system. (Krämer and Afchine, 2004) developed another expression (referred as K&A) for $0.007< U/U_0 <0.2$ based on computational fluid dynamics. However, for low particle sizes, the efficiency doesn't converge to 1. As a consequence, we have used the LZK (Liu et al., 1989) function since it covers most of the $U/U_0$ ratios we get in the inlet system, it agrees with the experimental data in Belyaev and Levin (1972) and Belyaev and Levin (1974) and it converges to $U_0/U$ for large particles sizes and 1 for small particle sizes. Outside its valid range ($U/U_0 < 0.1$), the LZK function agrees with the K&A function for large radius and converges to 1 for small particle sizes. The equation is valid for $0.01<Stks<100$, which is enough to cover the range in between 1 and 100 μm. As already stated, it tends to 1 for small particles sizes and to $U_0/U$ for large particles sizes (At 50 L min$^{-1}$, the ratio $U/U_0$ is 0.2). All the calculations were done under standard conditions (0 °C and 1 bar).

*Inlet inertial deposition is* defined as the inertial loss of aerosol particles when they enter nozzle. It is produced by the fact that the streamlines bend towards the walls at the moment they enter the nozzle, some large inertia particles can impact the walls and get deposited. Here, we have used the equation given in Liu et al. (1989) which quantifies this effect. It is also valid for $0.01<Stks<100$, which is enough to cover the range in between 1 and 100 μm.





*Turbulent inertial deposition* happens when some particles are collected by the wall when travelling in a pipe in the turbulent regime because some of the particles cannot follow the eddies of the turbulent flow. In order to include this mechanism, we used the equation given in Brockmann (2011), using the relation in between the deposition velocity and dimensionless particle relaxation time given by Liu and Ilori (1974). These calculations are valid for a cylindrical pipe, whereas the turbulent section of the inlet considered here is the nozzle, which has a conical shape. In order to account for this, we divided the conical nozzle into 90 conical sections with an increasing diameter and a length of 1mm, and combined the effect of all the sections. As already mentioned, above 65 L min⁻¹, turbulent flow occurs in the whole inlet tube. This has been taken into account in the 80 L min⁻¹ case in Fig. 2b. The equation used here has been tested for size ranges in between 1.4 and 20 µm, and doesn't depend on the Reynolds number values it was tested for (10000 and 50000) (Liu and Ilori, 1974).

*Bending inertial deposition* was also considered, since the line curves with an angle of 45° in order to bring the airstream into the cabin. The inertia of some particles may keep them in their original track and they are not able to follow the air streamlines that are bending towards the cabin, following the inlet tubes. In order to account for these losses, we have used the empirical equation given in Brockmann (2011) based on the data from Pui et al. (1987) for laminar flow. This equation was developed for Reynold numbers of 1000, and we have used it for higher values. However, in Brockmann (2011), one can see that the data from Pui et al. (1987) for Re=6000 (beginning of the turbulent flow regime) doesn't differ that much from the fit we have used (valid for Re=1000). Since our Re numbers for the thick section of the tube almost never go above 5000, we can still use the laminar flow fit. This model has been tested for 0.08 < Stks < 1.2, which is enough to cover most of the range where the inertial deposition efficiency drops from 1 to 0. The main caveat of this calculation is that the model considers that the flow rate before and after the bending is the same, while in the inlet system, if the bypass flow is on, the flow rate before and after the bending is different (before it, it would be equal to the total flow rate, whereas after the bending, it would be equal to the filter flow rate). As a consequence we assumed that the flow rate after the bending is equal to the total flow rate.

*Gravitational settling* was also considered. We used the analytical equation given by Thomas (1958), as stated in Brockmann (2011). We applied this equation for the section of the pipe from the nozzle to the bend (15 cm long). We used the modification (also analytical) of the previous equation given in Heyder and Gebhart (1977) in order to account for the losses in the second section of the tube which is 40 cm long and it is bended 45°. The gravitational losses in the nozzle were neglected since the settling distance is much shorter and the time the air takes to pass it is smaller since it travels quicker. As stated previously, the lower part of the turbulent regime can be reached for high flow rates through all the tube. For these cases, we still use this equation which is only valid for the laminar regime, since the gravitational settling efficiencies for the turbulent regime are very close to the laminar regime ones (Brockmann, 2011) and wouldn't make a significant difference in our calculations.

*Diffusional efficiency* accounts for the fact that small aerosol particles could diffuse to the walls of the pipe via Brownian motion. In order to account for this phenomenon, we have used the analytic equation by Gormley and Kennedy (1948) as stated in Brockmann (2011). We have assumed that diffusion happens only in the tube (before and after the bend) and excluding the diffusion in the nozzle since it is negligible because these losses are a function of the residence time and the residence time of the aerosol particles in the nozzle is much smaller than the rest of the tube. For this calculation, we have assumed 0 °C and 1 atm. We didn't show the efficiency associated to diffusion in Fig. 2a because it was very close to 1 for all considered sizes. It only becomes slightly smaller than 1 for sizes below 20 nm at 50 L min⁻¹. As a consequence, the inlet could be potentially used to sample nucleation





mode aerosol particles, even though for this study we will only focus on the particles larger than 0.1
µm.
*Filter collection efficiency* accounts for the fact that some particles can pass through the pores of the
filter, if they are smaller than the pores. However, filter pore size (in the case of polycarbonate
capillarity filters) and filter equivalent pore size (in the case of PTFE porous filters) is sometimes
misunderstood as a size cut off at which smaller particles are lost and larger particles are captured.
However, particle collection on filters happens through several mechanisms including interception,
impaction, diffusion, gravitational settling or by electrostatic attraction under certain conditions
(Flagan and Seinfeld, 1988; Lee and Ramamurthi, 1993). As a consequence, particles with diameters
below the pore size are normally collected (Lindsley, 2016; Soo et al., 2016). 99.48% of the generated
sodium chloride particles with sizes in between 10.4 and 412 nm were collected by a 0.4 µm
polycarbonate filter at flow rates below 11.2 L min$^{-1}$ (smaller than most of the flow rates at which the
air passes through the same filters in the FAAM filter inlet system) (Soo et al., 2016). As a consequence,
we assumed a filter collection efficiency of 100% across the whole considered size range (0.1 to 100
µm).
*Anisoaxial losses* have not been considered in the analysis shown in Fig. 2, after estimating that they
would only affect particles significantly larger than 10 µm and the fact that the alignment of the inlet
is difficult to quantify and the angle of attack changes during the flight. Using the equations explained
in Hangal and Willeke (1990a), we calculated that the modification of the sub-isokinetic behaviour of
the inlet produced by small values of θ is negligible. The equation was used beyond its experimental
limit, but this extrapolation was justified by the fact that the equation for θ = 0 made asymptotic
physical sense at the low and high Stokes number limits and produced very similar results to the ones
showed in Fig. 2a. Anisoaxial sampling can also produce inertial losses when particles impact the walls
of the inlet. These ones have been quantified using the expression given by Hangal and Willeke
(1990b) for different values of θ and they can be seen in Fig. 3. This mechanism looks very similar to
the gravitational and bend deposition efficiency shown in Fig. 2a. Anisoaxial inertial losses add a cut
off that prevents large particles to be sampled. As one can see in Fig. 3, the effect is very dependent
on the angle and only affects particles significantly larger than 10 µm in most cases, so it hasn't been
included in the total analysis shown in the Fig 2. One can see in Fig. 3 that the position of the D50 of
the anisoaxial cut off decreases when increasing values of θ up to 2°. For values of θ between 2° and
6°, it increases when increasing θ.

**Acknowledgements**
We a grateful for helpful discussions with Paola Formenti (Centre National de la Recherche
Scientifique), Hannah Price (FAAM) and Eduardo Morgado (School of Earth and Environment,
University of Leeds). Airborne data were obtained using the BAe-146-301 Atmospheric Research
Aircraft operated by Directflight Ltd (now Airtask) and managed by FAAM, which is a joint entity of the
Natural Environment Research Council (NERC) and the UK Met Office. We acknowledge the dedicated
work of FAAM, Directflight and Avalon. We thank all the people involved in the EMERGE, VANAHEIM
and MACSSIMIZE campaigns. We acknowledge the Leeds Electron Microscopy and Spectroscopy
Centre for the use of their microscopy facilities and the Centre for Environmental Data Analysis for
providing us with the FAAM datasets used here. We are grateful for funding from the European
Research Council (ERC) (648661 MarineIce).



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



**901    Figures**

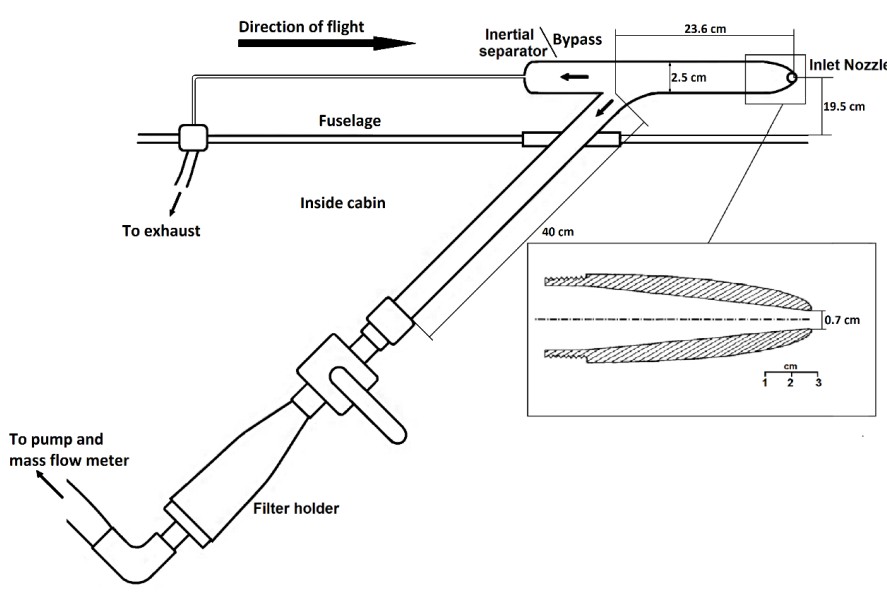


Fig. 1. Schematic diagram of one of the two parallel lines of the inlet system.



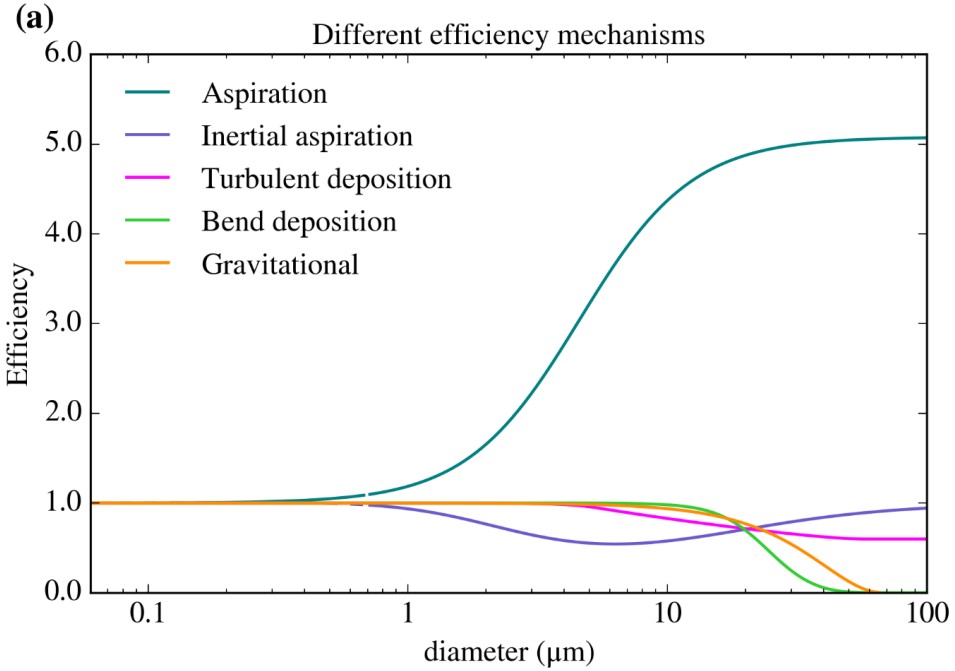


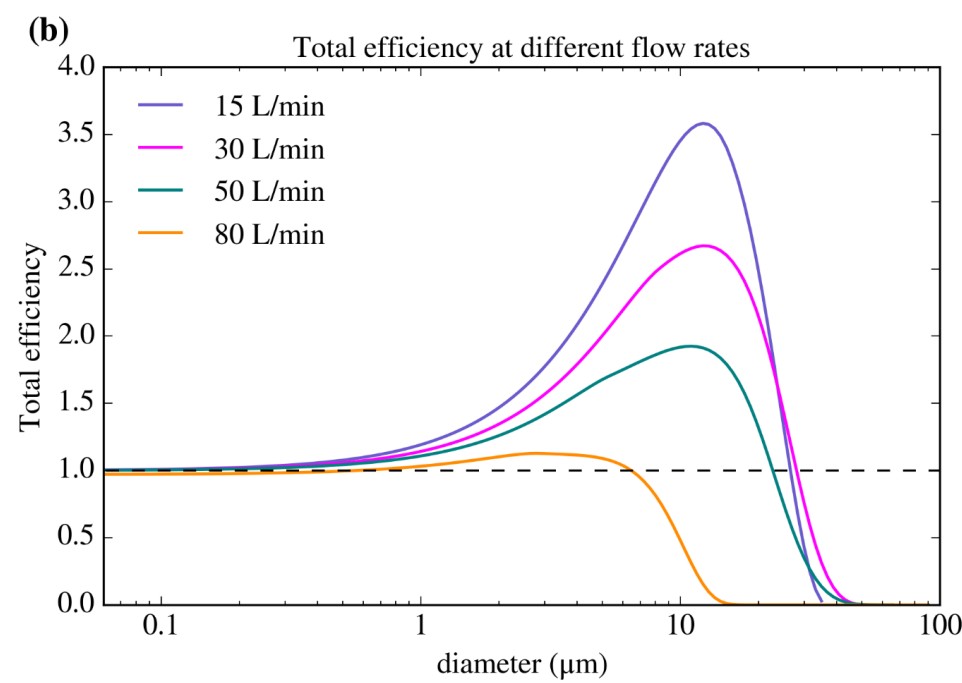





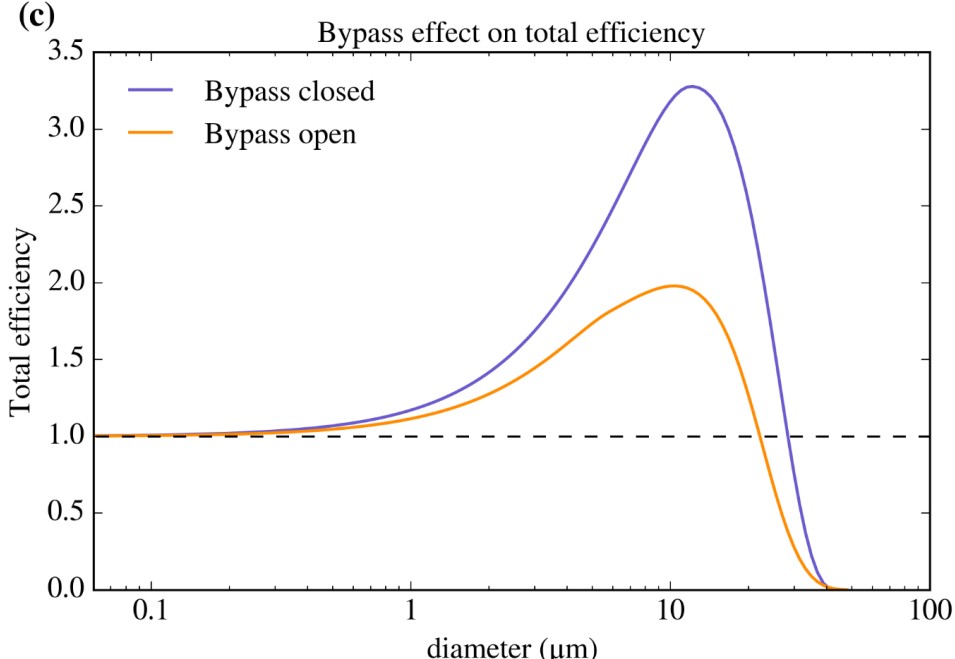


Fig. 2. *Theoretical efficiencies of the Filter inlet system.* (a) Efficiencies of the four mechanisms considered in this work for a total flow rate of 50 L min$^{-1}$. We have assumed a dynamic viscosity of 1.82x10$^{-5}$ kg m$^{-1}$ s$^{-1}$ (value for 0 °C) and a particle density of 1000 kg m$^{-3}$. The speed of the air mass (U$_0$) was 110 m s$^{-1}$, a typical FAAM flying speed at low altitudes. (b) Total efficiency for four different total flow rates. For the 80 L min$^{-1}$ case, turbulent deposition through the whole line was considered since the flow was turbulent through the whole pipe. (c) Total efficiency considering all the described mechanisms for a 20 L min$^{-1}$ filter flow rate with the bypass closed and a 20 L min$^{-1}$ filter flow rate with the bypass open (considering a bypass flow of 25 L min$^{-1}$).





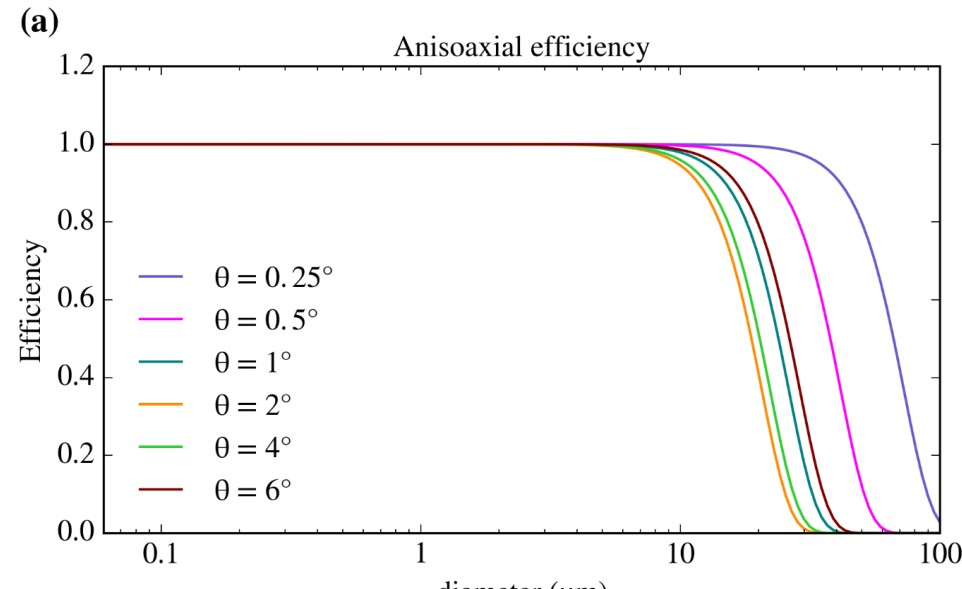

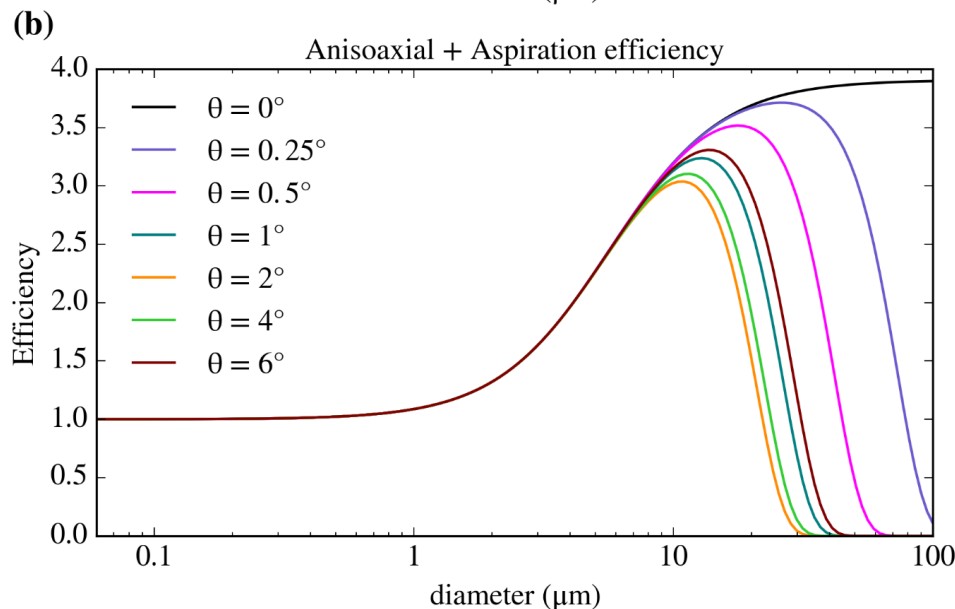

Fig. 3. Anisoaxial inertial losses of the sampling carried out by the Filters inlet system for different values of the angle in
between the inlet and the flight direction. The calculations have been presented by themselves (a) and combined with the
aspiration efficiency (b), which one can see in Fig. 2a. The anisoaxial calculations have been done using the equations given
by (Hangal and Willeke, 1990b), using the same parameters and dimensions than in Fig. 2, apart from the flow rate, which
was set to 65 L min$^{-1}$ in order to be within the valid range of $U/U_0$ that was used to develop the equation. For smaller or
larger values of the flow rate (under which most of the sampling is carried out), the differences in the efficiency from the
ones show here are minimal.






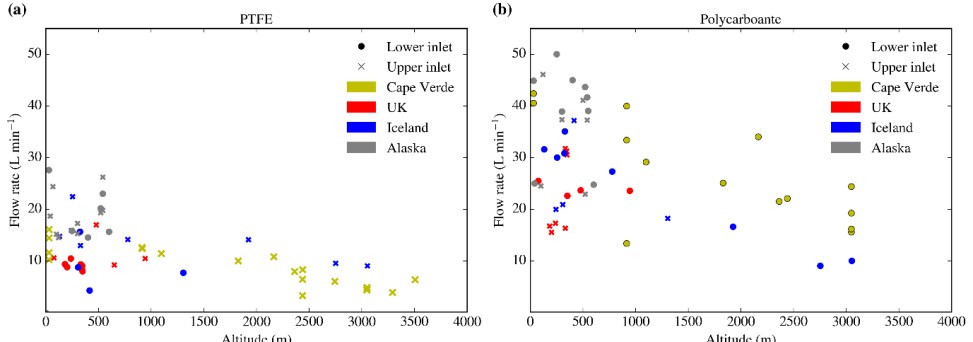



Figure 4. Filter flow rate of different samplings carried out in different campaigns at each altitude using: (a) Sartorius PTFE membrane filters (47mm diameter with a pore size of 0.45μm) and (b) Whatman nucleopore polycarbonate track etched filters (47mm diameter with a pore size of 0.4μm). The crosses represent samples taken in the upper line of the inlet system, whereas dots represent the sampling in the bottom line. Different mesh supports were used for the data collection. The data from Cape Verde was extracted from (Price et al., 2018) and the notes of the analysis carried out by the authors whereas the altitude data from the other three was obtained from the pressure altitude measurement carried out by the Reduced Vertical Separation Minimum system on board of the aircraft. The FAAM core datasets used (via the Centre for Environmental Data Analysis) were C019, C022, C024, C025, C058, C059, C060, C061, C062, C063, C085, C086, C087, C088, C089, C090 and C091. The bypass was closed for all the data in Cape Verde whereas it was open for all the data in the other campaigns. Note that the flow rate here corresponds to the filter flow rate (measured with the mass flow meter), not the total one.




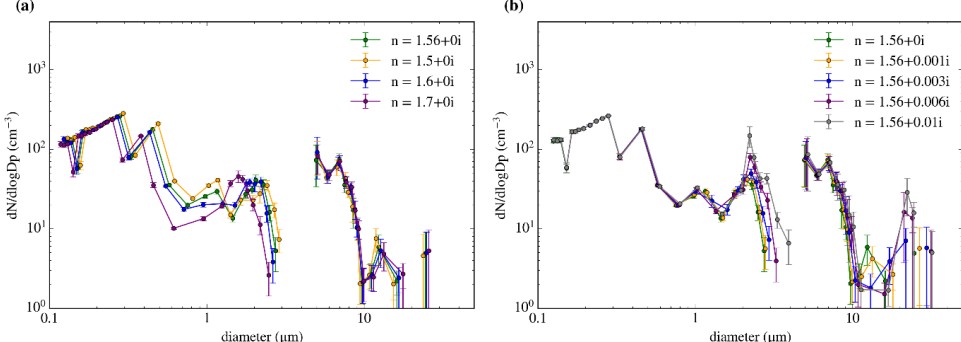


Figure 5. Sensitivity of the size distributions measured by the PCASP-CDP during the C010 flight on the 2017/05/10 from
11:24 to 11:38 UTC to small variations in the refractive index. We tested both the real part (a) and imaginary part (b). The
errors are calculated according to the methods explained in Sect. 3.






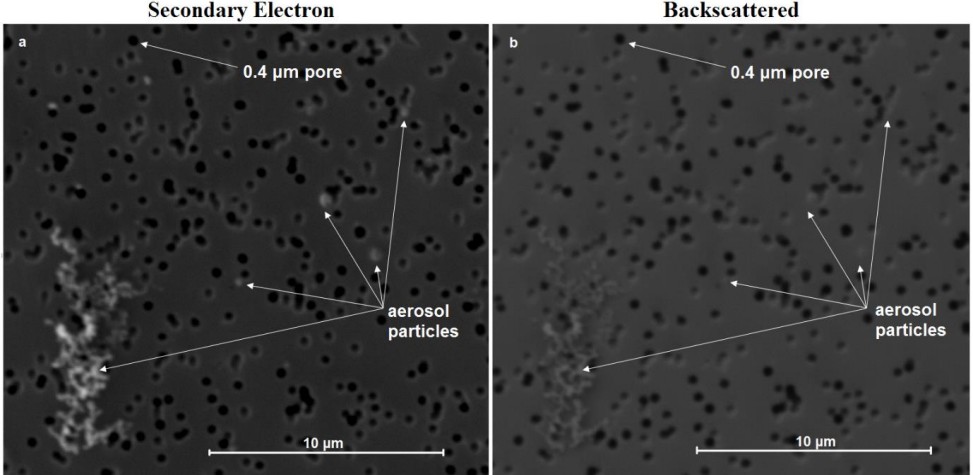


Figure 6. Secondary electron image (a) and Back Scattered Electron image (b) of the same area of the same filter, collected
on the 2018/07/05 from 13:32 to 13:47 in the upper line with the bypass open. As one can see, some of the small particles
in the SE image appear almost transparent under the BSE image. Even the 10µm soot particle in the bottom left of the
image shows a very low contrast in the BSE image.






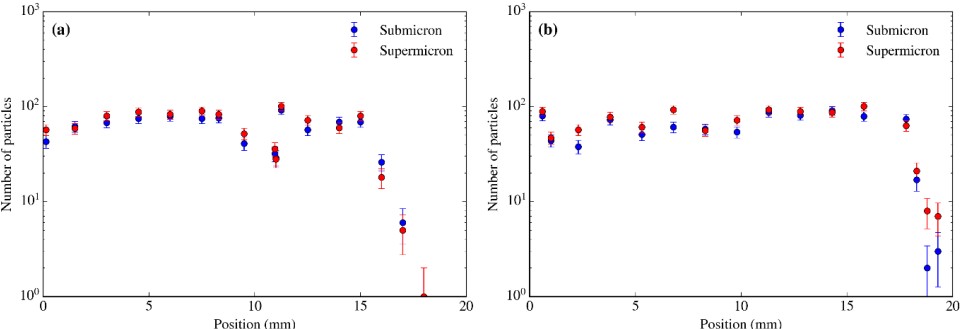


Figure. 7. Radial distribution of particles test on the sample collected on the 2017/10/02 from 16:24 to 16:40 UTC about 320
m high in south Iceland, using the lower line and open bypass, sampling 219 L. Number of submicron and supermicron
particles in same size areas (~160x190 μm²) radially distributed versus the distance from the approximate centre through a
radius of the filter (a) and another trajectory from the centre of the filter deviated 30° from the first radius (b). The analysis
was done at 20 KeV and x5000. The number of both supermicron and submicron particles remains very constant all over the
surface of the filter, until reaching the edges of it (which are cover by a rubber O-ring during the sample) and the number of
particles drops to the limit of the detection within a few millimetres. The error in the number of particles comes from Poisson
counting statistics.






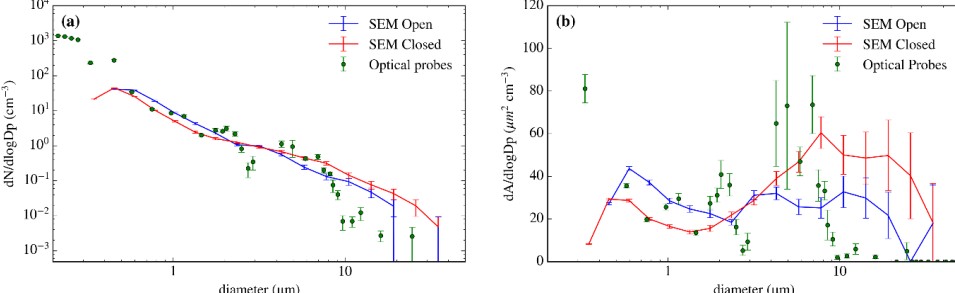

Figure 8. Bypass test carried out during the C010 flight on the 2017/05/10 from 11:24 to 11:38 UTC. The lower line sampled
226 L with the bypass closed, whereas the upper line sampled 141 L with the bypass open. The flow rates were 16.1 L min⁻¹
and 10.6 L min⁻¹ respectively. The optical probes are the PCASP-CDP, using the closest calibration to the sampling date and a
refractive index of 1.56 as stated in the Sect. 2.3. The data is shown in both number size distribution (a) and surface area size
distribution (b). The only error source considered for the SEM size distribution is the Poisson counting error, which has been
included this figure and all the subsequent figures including SEM size distributions.



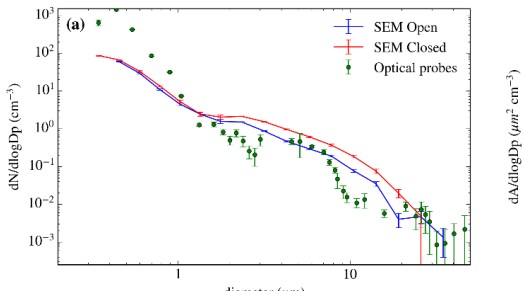 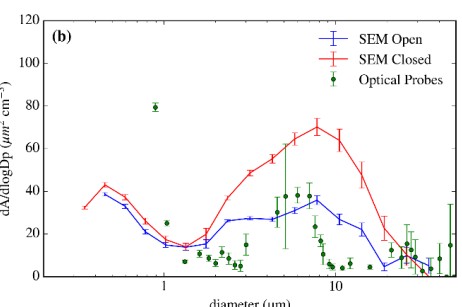

Figure 9. Bypass test carried out during the C057 flight on the 2017/09/27 from 13:33 to 13:50 UTC. The lower line sampled 555 L with the bypass open, whereas the upper line sampled 499 L with the bypass closed. The flow rates were 34.7 L min$^{-1}$ and 31.2 L min$^{-1}$ respectively. The position of the closed and open line was swapped with respect to the first analysis in Fig. 8. The optical probes are the PCASP-CDP, using the closest calibration to the sampling date and a refractive index of 1.56 as stated in the Sect. 2.3. The sampling was interrupted for a minute to avoid a turn.



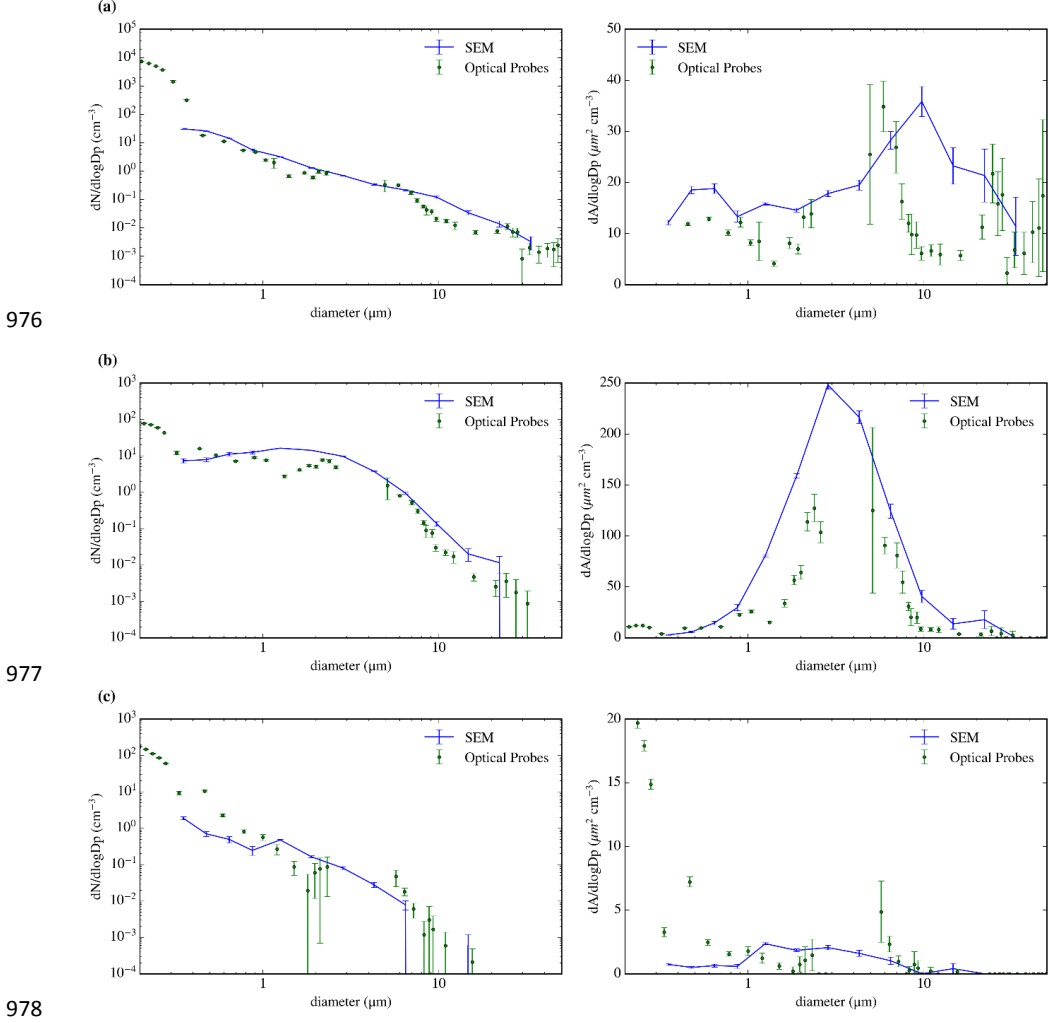

976

977

978

Figure 10. SEM obtained size distribution compared with PCASP-CDP online size distribution for three different sampling periods in three different aerosol environments. Close to London, on the 2017/07/19 from 15:20 to 15:51 UTC, sampling 953 L (a), south of Iceland on the 2017/10/02 from 16:24 to 16:40 UTC, sampling 432 L (b) and in north Alaska on the 2018/03/20 from 20:15 to 20:37, sampling 724 L (c). All the sampling was done in the upper line with the bypass open. The flow rates through the filter holders are 30.9, 30.5 and 42.0 L min[-1] respectively. The optical probes are the PCASP-CDP, using the closest calibration to the sampling date and a refractive index of 1.56 as stated in Sect. 2.3.






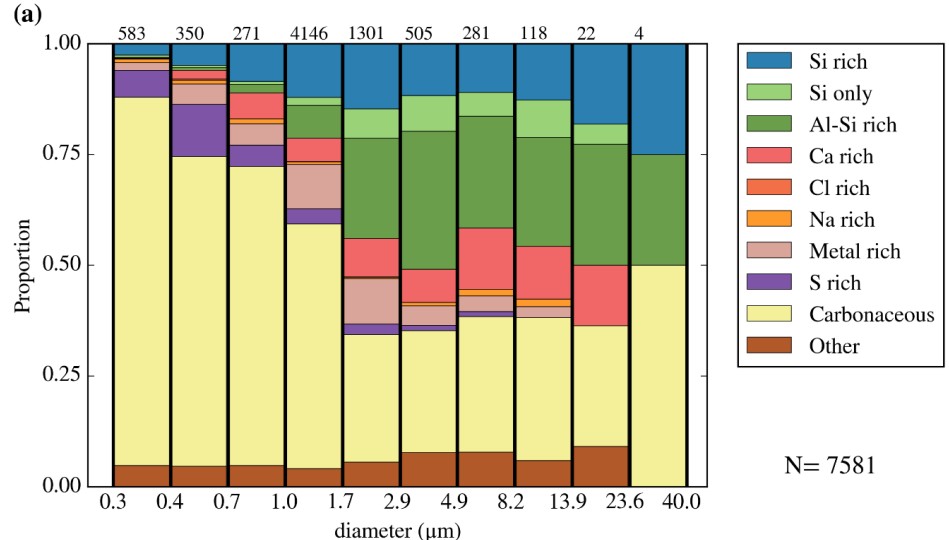







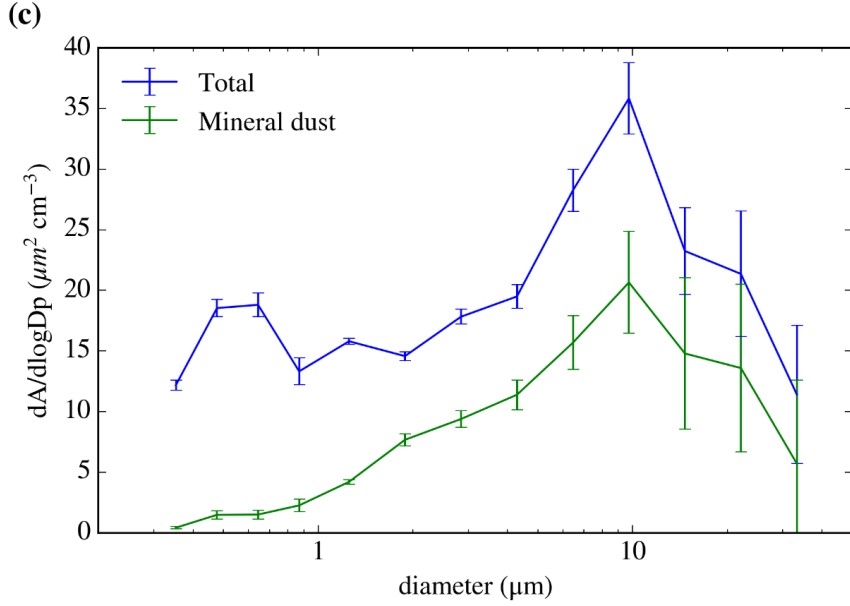


Fig. 11. Size-segregated compositional and morphological analysis of a sample collected close to London on the 2017/07/19
from 15:20 to 15:52 UTC by the lower line with the bypass open, sampling a total of 953L. (a) Fraction of particles
corresponding to each compositional category (described in the Sect. 7) for each size. The number of particles per bin can
be seen in the top of the figure. (b) Number size distribution for each composition. Cl rich particles were not included since
only two particles in this category were found. The errors have been calculated from the Poisson counting statistics (applying
it to both the size distribution and the compositional measurements). (c) Surface area of both all the detected aerosol
particles and the ones whose composition was consistent with mineral dust. Errors have been calculated in the same way as
before. By integrating the green curve in the figure (c) we obtained the total surface area of mineral dust in the sample (19.1
$\mu m^2$ cm$^{-3}$).

998