# Peer review of "Characterisation of the filter inlet system on the BAE-146 research aircraft and its use for size resolved aerosol composition measurements"

_Atmospheric Measurement Techniques, 2019_

## Referee Comment (RC1) · Konrad Kandler (Referee) · 12 Jun 2019

The manuscript of Sanchez-Marroquin et al. deals with the characterization of an aircraft inlet frequently used in the British BAE-146 research aircraft. Despite of the importance of inlet characterization, it happens frequently that aerosol inlets are built and used, but remain uncharacterized. Therefore, these type of studies is valuable to rate the results of aerosol research done with the according systems, in particular with respect to their (size) representativity.

The authors compare an experimental approach for inlet transmission characterization with a theory-based one and come to the conclusion of a general approximate agreement. They propose a range of operational conditions based on their results.

The paper is mostly well-written; the methods are explained and applied. Some unclear sections remain (detailed below). References are adequate. However, some effort should be placed into the thermodynamic considerations, and the SEM part should be structured partly into a second publication. Also, some intentions for future work are given scattered through the paper, which should be either moved to the motivation section or omitted.

General remarks

The paper goes into details about aerosol flows, but the properties and values reported in the text should be treated with more precision. E.g., flow rates are reported in L per minute, but it is unclear, whether this means volumetric L at the outside conditions, volumetric L at the inlet conditions, mass equivalent L at standard conditions.

636-637: "All calculation were done under standard conditions" – Why? Most aerosol/carrier gas interactions depend on the air viscosity and free mean path, some on Reynolds number and therefore density. As result, most efficiency functions at the end have temperature and density in them. It doesn't seem to be a wise choice to neglect these dependencies, in particular not for aircraft measurements with their strong variation. Also, the thermodynamic conditions change considerably from the outside conditions through in inlet and tubing to the filter.

Estimates were done using 'classical' aerosol aspiration / transmission formulas, which don't appear to be relevant in all cases (see detailed comments). A major question in this context is why the authors decided not to use computational fluid dynamics modeling. While these techniques are work-intensive and in turbulent situations also not necessarily precise, in particular for the inlet, diffusor and bend / inertial separator section, they might have been more useful.

The section 8 and 9 appear as a misfit in the context of this technical paper. I suggest removing it here and extending it into a standalone paper or letter in another journal. A proof of capability of SEM and measurements with the filter system onboard the aircraft is not really required, as this has been done during decades (Johnson et al., 1991; Formenti et al., 2003; Chou et al., 2008; Johnson et al., 2012). In addition, the plots are shown, but remain un-discussed, and no context (e.g., meteorology, trajectories, campaign aims) is given. If anything of the SEM compositional results should be included, I suggest including the sensitivity tests for the classification scheme as function of the detection limit (currently in the supplement), as from this you can derive recommendations with respect to element quantification settings.

Detailed remarks / corrections

Abstract 28-30: While this is surely true, it's not part of the paper.

40 Missing ".".

55 "... has been limited." was not carried out?

57-67: These lines are more a summary than an introduction. As it is partly redundant to section 9, I suggest removing it.

88: It doesn't get clear from the picture: does the 0.7 cm inlet have the inner edges rounded? From the references literature I would think that it is.

97: The numbers indicate a high precision, which is usually not achieved by mass flow meters (1 – 3 % uncertainty). How water vapor was treated, which influences the reading?

98: It probably reports the gas mass, not volume.

101: Rietschle?

110: Particles are not necessarily lost (to the wall), but can be diluted (i.e. not entering the inlet).

112-113: and depending on pressure and temperature...?

122: It seems that to rate the importance of a mechanism, its effect needs to be compared to all others. Was this done, or were the only most probably important mechanisms selected? Please explain the reasoning.

126-162: Too redundant with the appendix. Suggestion: either refer to the appendix and remove all short explanations here, or include the full discussion currently in the appendix. For a technical journal, also the latter would be appropriate.

181-182: How does the bypass change the temperature in the inlet (probably mitigate heating by less deceleration)? Is the effect strong enough to impact on volatile particles?

213: It is somewhat surprising that the filter flow appears to be unregulated. Maybe, a regulation system should be included in future as well.

217: "microscope"

244: "highly unlikely" instead of "not likely"?

262: "regarded" instead of "shown"?

263: Regarding the "reference": just recently, there was a publication showing size distribution distortion for the 'free-stream' instruments, too, (Spanu et al., 2019), which might be worth checking.

281: Kandler et al. used mostly backscatter electron, except for small particles on TEM grids. Check also the other references please.

284: Is it possible to quantify the undercounting of backscatter versus secondary electron? That might be valuable information for people dealing with similar questions.

283-298: Can you include an image showing the benefit of the Ir coating and the potential size increase? Again that appears to be valuable information.

301-304: it appears to be more meaningful to specify the pixel size in nm (scanning grid size), instead of the magnifications, which are screen-related.

310: Was the ECD converted into aerodynamic or optical equivalent diameter or just used "as it is"? Please discuss, as this might introduce certain biases.

316: "evenly": In Fig. 7, a min/max variation of a factor of three is visible, interestingly without a size bias. Was it the same in all radial directions, or is that random fluctuations?

320: Please 'link' "ECD" to "equivalent circular diameter".

322-323: Where these charging problems observed despite the relative thick Ir coating?

414-415: I suggest treating this more precisely. At 10 $\mu$m, there is a disagreement of about a factor of 10 or slightly more, and the theory predicts between 2 and 5. Considering the uncertainties, it's probably fair to call this agreement. At 2 $\mu$m, there is the same factor > 10 difference, but the theory says 1. Here, 'agreement' becomes stretched. However, the optical particle counter curves have persistent minima (3 $\mu$m, 10 $\mu$m) and maxima (2 $\mu$m, 5 $\mu$m), where the SEM curve is smooth. Are these minima/maxima realistic or potentially an artifact of a failing Mie inversion?

479: "sulphate aerosol particles, which are solid or liquid sulphuric acid particles" If it is sulphate (probable), then it is a salt as reaction product from an acid with something else. Solid sulphuric acid on a filter is improbable. Please correct the phrasing. Also, particles in this category could be organo-sulphates.

501: "chloride". Potassium-rich Cl- (and/or S-)containing particles are known from biomass burning (Li et al., 2003; Lieke et al., 2011), and other Cl-rich from (waste) incineration (Willison et al., 1989; Graedel and Keene, 1995).

512: How about fractionated crystallization of a sea-water droplet on the filter, leading to separate NaCl, MgCl, CaCO3 or CaSO4 particles?

648-650: This approach appears to be questionable, as turbulence for an increasing diameter tube probably has an additional generation mechanism (inertia), compared to turbulence in a constant diameter tube (mostly by shear). Please comment.

654-669: The bend approximation assumes a smooth tube, too. If it was used for the droplet separator, the conditions are not met. Also, if the flow is decelerated during the bend, large particles might become accumulated on the outer side, which is not accounted for by the simple approximation. Please discuss.

683: For diffusion a constant diameter bend can probably be well-approximated by a straight tube.

691-703: While it is correct that the particles are retained by the filter, not necessarily all particle sizes can be analyzed by microscopy techniques (representatively), as the smaller particles might be deposited inside the pores, too.

707: The referred equations apply to sharp-edged nozzles, while in the setup blunt and probably rounded ones are used (according to the aircraft engine inlet description). In particular the inlet rounding is done to mitigate misalignment effects (Hermann et al., 2001).

924: Caption "Polycarbonate". As many effects discussion above might be closer related to the volumetric flow rate than to the mass flow rate, it should be shown in addition. The Iceland/Cape Verde ratio is inverted for the two filter types or two inlet types. How can this be explained?

Chou, C., Formenti, P., Maille, M., Ausset, P., Helas, G., Harrison, M., and Osborne, S.: Size distribution, shape, and composition of mineral dust aerosols collected during the African Monsoon Multidisciplinary Analysis Special Observation Period 0: Dust and Biomass-Burning Experiment field campaign in Niger, January 2006, J. Geophys. Res., 113, D00C10, doi: 10.1029/2008JD009897, 2008.

Formenti, P., Elbert, W., Maenhaut, W., Haywood, J., and Andreae, M. O.: Chemical

composition of mineral dust aerosol during the Saharan Dust Experiment (SHADE) airborne campaign in the Cape Verde region, September 2000, J. Geophys. Res., 108, 8576, doi: 10.1029/2002JD002648, 2003.

Graedel, T. E., and Keene, W. C.: Tropospheric budget of reactive chlorine, Global Biogeochemical Cycles, 9, 47-77, doi: 10.1029/94gb03103, 1995.

Hermann, M., Stratmann, F., Wilck, M., and Wiedensohler, A.: Sampling Characteristics of an Aircraft-Borne Aerosol Inlet System, J. Atmos. Ocean. Tech., 18, 7-19, doi: 10.1175/1520-0426(2001)018, 2001.

Johnson, B., Turnbull, K., Brown, P., Burgess, R., Dorsey, J., Baran, A. J., Webster, H., Haywood, J., Cotton, R., Ulanowski, Z., Hesse, E., Woolley, A., and Rosenberg, P.: In situ observations of volcanic ash clouds from the FAAM aircraft during the eruption of Eyjafjallajökull in 2010, Journal of Geophysical Research: Atmospheres, 117, doi: 10.1029/2011jd016760, 2012.

Johnson, D. W., Kilsby, C. G., McKenna, D. S., Saunders, R. W., Jenkins, G. J., Smith, F. B., and Foot, J. S.: Airborne observations of the physical and chemical characteristics of the Kuwait oil smoke plume, Nature, 353, 617-621, doi: 10.1038/353617a0, 1991.

Li, J., Pósfai, M., Hobbs, P. V., and Buseck, P. R.: Individual aerosol particles from biomass burning in southern Africa: 2, Compositions and aging of inorganic particles, J. Geophys. Res., 108, D8484, doi: 10.1029/2002JD002310, 2003.

Lieke, K., Kandler, K., Scheuvens, D., Emmel, C., Von Glahn, C., Petzold, A., Weinzierl, B., Veira, A., Ebert, M., Weinbruch, S., and Schütz, L.: Particle chemical properties in the vertical column based on aircraft observations in the vicinity of Cape Verde Islands, Tellus, 63B, 497-511, doi: 10.1111/j.1600-0889.2011.00553.x, 2011.

Spanu, A., Dollner, M., Gasteiger, J., Bui, T. P., and Weinzierl, B.: Flow-induced errors in airborne in-situ measurements of aerosols and clouds, Atmos. Meas. Tech.

Discuss., 2019, 1-46, doi: 10.5194/amt-2019-27, 2019.

Willison, M. J., Clarke, A. G., and Zeki, E. M.: Chloride aerosols in central northern England, Atmos. Environ., 23, 2231-2239, doi: 10.1016/0004-6981(89)90185-6, 1989.

---

## Referee Comment (RC2) · Anonymous Referee #2 · 10 Jul 2019

This paper presents a characterization of the filter inlet system of the research aircraft BAe146. It includes calculated inlet sampling and transmission efficiency, a description of the analysis of the filter samples by scanning electron microscopy (SEM), and a comparison of the size distributions obtained by SEM with underwing aerosol and cloud probes.

Unfortunately, the manuscript suffers from being vague at important points. Especially for a technical journal, a comparison between calculations and measurements needs to discussed in more detail. Also, expressions like "in agreement" are used frequently

where a precise numbers (with error limits) would have been necessary. Thus, I can not recommend publication in the current stage and suggest some major revision before publication:

Major points:

As said above, the manuscript lacks precise numbers. Many statements are vague, like "in agreement" or "minor fraction" etc. This is not sufficient for a technical journal.

Furthermore, the SEM part is a description of the classification, but no further validation is done. Additional aircraft-based gas (e.g. CO) and particle measurements (mass spectrometers?) may help to characterize the air mass origin and the particle properties and thereby validate the composition. The comparison of SEM size distribution with the PMS probes is not very conclusive, because only qualitative statements ("in good agreement") are made.

Furthermore, the size distributions of the PCASP (Fig. 5) seem to have a problem at 300 nm and above 2 $\mu$m. The PCASP shows decreasing number concentrations above 2 $\mu$m while the CDP starts at 5 $\mu$m with much higher number concentrations. Does the PCASP underestimate particle number above 2 $\mu$m? If so, would it be better to omit these points and use a lognormal fit to the reliable CDP and PCASP data to obtain realistic fine and coarse mode distributions? To what extend can such size distributions validate the inlet efficiency if the uncertainties are so high?

Figure 8-10: Have the SEM data been corrected for the calculated inlet transmission and aspiration efficiency? I could not find a statement on this in the text. If not, then an overestimation of about a factor 3 - 4 around 10 $\mu$m should be observed (from Fig 3b). Is that the case? By bare eye, the factor seems to be larger than three, but there is no discussion in the text, except for a "good agreement" statement.

Minor

I was a bit confused by the mixture of sampling efficiency study and chemical compo-

sition study. I see that both needs to be done, but I needed some time to realize that the manuscript focuses on these two topics. Mabye a change of the title would help the reader.

Specific comments

Line 353: "This happens more frequently for smaller particles, but it can also happen with some larger particles..." What is "smaller" and "larger" here? Please be more precise and give a size range.

Line 366-368: "The number of particles is very low, typically about the order of magnitude of one particle per 100 by 100 $\mu$m square, which is well below the typical particle loading on a filter exposed to the atmosphere" Please give numbers for typical particle loading. "Well below" is not quantitative.

Line 373: "...from the analysis of atmospheric aerosol (it was only ever a very minor component)." Please specify "very minor"

Line 374: "By doing this, we make sure that we excluded more than half of the artefacts of the analysis" I don't understand. Before that, you said that >90% contained Cr, so you would remove >90 of the artifact, isn't it?

Section 7 Did you observe any signs of meteoric material (see Murphy et al., 2014)? Particles dominated by Fe, Mg, Si and S ?

Line 501: "sodium chlorine" -> sodium chloride

Fig 4, caption: "FAAM core datasets" have not been explained before

Fig 5 + lines 257-264: As already written above, the size distributions of the PCASP (Fig. 5) seem to have a problem at 300 nm and above 2 $\mu$m. The PCASP shows decreasing number concentrations above 2 $\mu$m while the CDP starts at 5 $\mu$m with much higher number concentrations. Does the PCASP underestimate particle number above 2 $\mu$m? If so, would it be better to omit these points and use a lognormal fit to the

reliable CDP and PCASP data to obtain realistic fine and coarse mode distributions? What happens at 10 $\mu$m with the CDP?

Figs 8 and 9: I suggest combining Figs 8 and 9 into one figure with 4 graphs

Fig 8, 9, 10 and line 415: "The results of these comparisons are in agreement with the theoretical calculations in Sect. 2.2." Did you correct the SEM size distribution with the calculated sampling efficiency? Can you divide SEM dN / PMS dN and derive an "experimental" sampling efficiency and compare that to the calculated curves in Sect. 2.2? One of the above should be done, otherwise your statement "are in agreement" is too weak.
* * *

---

## Author Comment (AC1) · 12 Sep 2019

The comment was uploaded in the form of a supplement:
https://www.atmos-meas-tech-discuss.net/amt-2019-196/amt-2019-196-AC1-supplement.pdf

---

## Author Comment (AC2) · 13 Sep 2019

The manuscript of Sanchez-Marroquin et al. deals with the characterization of an aircraft inlet frequently used in the British BAE-146 research aircraft. Despite of the importance of inlet characterization, it happens frequently that aerosol inlets are built and used, but remain uncharacterized. Therefore, these type of studies is valuable to rate the results of aerosol research done with the according systems, in particular with respect to their (size) representativity.

The authors compare an experimental approach for inlet transmission characterization with a theory-based one and come to the conclusion of a general approximate agreement. They propose a range of operational conditions based on their results.

The paper is mostly well-written; the methods are explained and applied. Some unclear sections remain (detailed below). References are adequate. However, some effort should be placed into the thermodynamic considerations, and the SEM part should be structured partly into a second publication. Also, some intentions for future work are given scattered through the paper, which should be either moved to the motivation section or omitted.

Thank you very much for this comments. They are very useful and will definitely improve the manuscript. We address the specific comments below.

General remarks
The paper goes into details about aerosol flows, but the properties and values reported in the text should be treated with more precision. E.g., flow rates are reported in L per minute, but it is unclear, whether this means volumetric L at the outside conditions, volumetric L at the inlet conditions, mass equivalent L at standard conditions.
Line Added in Sect. 2.1: "The air flow through the filter (filter flow) is measured by a mass flow meter, which measures the sampled air mass and reports it in equivalent litres at standard conditions (273.15 k, 1013.529 hPa)".
Line Added in Sect. 2.2: (volumetric L at standard conditions: 273.15 k, 1013.529 hPa),
Line Added in Sect. 2.2: (all the flow rates of our calculations are given in L min-1 at standard conditions: 273.15 k, 1013.529 hPa).

636-637: "All calculation were done under standard conditions" – Why? Most aerosol/carrier gas interactions depend on the air viscosity and free mean path, some on Reynolds number and therefore density. As result, most efficiency functions at the end have temperature and density in them. It doesn't seem to be a wise choice to neglect these dependencies, in particular not for aircraft measurements with their strong variation. Also, the thermodynamic conditions change considerably from the outside conditions through in inlet and tubing to the filter. Estimates were done using 'classical' aerosol aspiration / transmission formulas, which don't appear to be relevant in all cases (see detailed comments). A major question in this context is why the authors decided not to use computational fluid dynamics modelling. While these techniques are work-intensive and in turbulent situations also not necessarily precise, in particular for the inlet, diffusor and bend / inertial separator section, they might have been more useful.
A temperature and pressure dependence test was performed. In the figure below, one can see the total efficiency for the 40L/min case (including the diffusion) for standard conditions, International standard atmosphere conditions at 0 and 3000m (the range in which the filter inlet system works. The differences are negligible. A similar negligible dependence was calculated for another inlet on board of the FAAM aircraft in Trembath (2012).

[Figure]

Line added at the third paragraph of Appendix A: "The effect of changes in pressure and temperature (and therefore air density and dynamic viscosity) that normally occur in the filter inlet system sampling range (0 to 3000 m) are negligible in all the used equations".

Further thermodynamic analysis in order to estimate the heating losses are impossible to carry out because of the lack of temperature and pressure measurement instruments through the inlet, which are not possible to have because of certification issues. This is a frequent problem of aircraft research. This is now mentioned in section:

Added in fourth paragraph of Sect. 5: "Also, volatilization of certain type of aerosol particles (which are more abundant in the submicron fraction (Seinfeld and Pandis, 2006)) can happen during heating (in this case produced by deceleration of the flow in the inlet) or sampling…"

The authors are aware that CFD (if carried out properly) would be a good way to characterise the inlet system, but decided not to include any CFD because it is far beyond the scope of the PhD project of the main author and it is very unlikely that it would change the conclusions of the paper. In addition, we think that the use of the appropriate empirical equations in combination with the comparison with underwing optical probes is an effective means of determining the best method of using the inlet and what biases can be expected.

The section 8 and 9 appear as a misfit in the context of this technical paper. I suggest removing it here and extending it into a standalone paper or letter in another journal. A proof of capability of SEM and measurements with the filter system on board the aircraft is not really required, as this has been done during decades (Johnson et al.,1991; Formenti et al., 2003; Chou et al., 2008; Johnson et al., 2012). In addition, the plots are shown, but remain un-discussed, and no context (e.g., meteorology, trajectories, campaign aims) is given. If anything of the SEM compositional results should be included, I suggest including the sensitivity tests for the classification scheme as function of the detection limit (currently in the supplement), as from this you can derive recommendations with respect to element quantification settings.

We interpreted that this comments refers to the sections 7 and 8 (SEM compositional categories, and an example of an application).

Sensitivity tests have now been included in the main paper (Appendix C).
Sect. 7 (The Section regarding to the compositional categories) has been moved to the Appendix B, in order to make the paper flow better.

Sect. 8 (now Sect 6), which includes some examples of what the technique can do has been kept just as an example without further discussion since this is a techniques paper. Further publications including all the SEM data collected by the authors and its discussions are already in preparation, and they will refer to this publication, rather than describing the technique in multiples SI sections of these future publications. The SEM technique has been used by other parallel projects in ground collected samples, which will also refer to this work.
Add in Sect. 8: "The purpose of this section is purely to give examples of the capabilities of the technique, further analysis is planned for subsequent papers"

On the comment about the SEM technique being used in the past. Yes, it has been used in the past, but our approach draws on elements of a number of previous studies and the classification scheme is novel. It therefore needs to be described somewhere. Hence, we think this techniques paper is the perfect place to include this.

Detailed remarks / corrections
Abstract 28-30: While this is surely true, it's not part of the paper.
Removed

40 Missing "."
Corrected

55 "... has been limited." was not carried out?
Some efforts have been made, we reviewed these previously in Price et al. (2018), but they are very limited. We are more specific in section 8 where we refer to the relevant papers. We have added the relevant references to this statement in the introduction as well.

57-67: These lines are more a summary than an introduction. As it is partly redundant to section 9, I suggest removing it.
Yes, this paragraph is the summary of what will come in the paper. We usually structure our papers in this way since it helps focus in on the specific objectives of the paper.

88: It doesn't get clear from the picture: does the 0.7 cm inlet have the inner edges rounded? From the references literature I would think that it is.
The word curved has been added to the text, as done in the given references.

97: The numbers indicate a high precision, which is usually not achieved by mass flow meters (1 – 3 % uncertainty). How water vapour was treated, which influences the reading?
The uncertainty of the MFM has been added (See caption of Fig. 4). Since the error is 1 % of the full scale (and this one is 400 L/min), the errors are above 1%.

Water vapour was neglected, since its effect is negligible. The difference in the heat capacity of dry air and saturated air at 20 ºC is about 2%. The difference in the molar mass of dry air and saturated air at 25 ºC is about 1.2%.
Add in Sect. 2.2: "The presence of water vapour hasn't been corrected since its effect is negligible."

98: It probably reports the gas mass, not volume.
It measures the gas mass but reports the equivalent volume at standard conditions.

101: Rietschle?
Added in Sect. 2.1: Elmo Rietschle (Gardner Denver Inc.)

110: Particles are not necessarily lost (to the wall), but can be diluted (i.e. not entering the inlet).
It is not clear how aerosol would be diluted in this inlet.

112-113: and depending on pressure and temperature...?
Added to second paragraph of Sect. 2.2: "The sampling efficiency of any inlet depends on the flow rates, and the flow regime (laminar vs turbulent), the pressure and the temperature."

122: It seems that to rate the importance of a mechanism, its effect needs to be compared to all others. Was this done, or were the only most probably important mechanisms selected? Please explain the reasoning.

We have included and excluded the same mechanisms as described in von der Weiden et al. (2009) (the reference has been added to this line in the text).

Add in third paragraph of Sect. 2.2: (a discussion on the choice of equations, how they have been applied and the excluded mechanisms can be found in Appendix A)

Added in last paragraph of Appendix A: "Other losses: Some mechanisms (thermophoresis, diffusiophoresis, interception, coagulation and re-entrainment of deposited particles) have not been considered, since they are second order mechanisms under our conditions when compared with the calculated mechanisms (Brockmann, 2011; von der Weiden et al., 2009) and for one of them (electrostatic deposition) it is not possible to quantify them. Electrostatic deposition is normally avoided by using grounded conductive materials so no electrical field exists within the tubing (Brockmann, 2011). Since the FAAM BAe-146 research aircraft is not grounded during the flight, we cannot state this mechanism is irrelevant. However, the experimental agreement between the SEM and optical probes suggest that this is a minor loss mechnanism."

126-162: Too redundant with the appendix. Suggestion: either refer to the appendix and remove all short explanations here, or include the full discussion currently in the appendix. For a technical journal, also the latter would be appropriate.
Most of the explanations have been removed.

181-182: How does the bypass change the temperature in the inlet (probably mitigate heating by less deceleration)? Is the effect strong enough to impact on volatile particles?
The fact that it is not possible at all to have temperature and pressure measurements through the inlet system (for certification reasons) limits our understanding of the bypass system. We can only state qualitatively that, as you mention, they bypass will decrease heating trough less deceleration (and maybe removing some heat from the system).

213: It is somewhat surprising that the filter flow appears to be unregulated. Maybe, a regulation system should be included in future as well.
Yes, we are recommending this as a part of a mid-life upgrade of the FAAM aircraft.

217: "microscope"
Fixed

244: "highly unlikely" instead of "not likely"?
Added
262: "regarded" instead of "shown"?
Added

263: Regarding the "reference": just recently, there was a publication showing size distribution distortion for the 'free-stream' instruments, too, (Spanu et al., 2019), which might be worth checking.
Although we are aware that the probes might have some sampling biases, as stated in Rosenberg et al. (2012), we still decided to use them as a refernece, as in previous works (Chou et al., 2008; Young et al., 2016; Ryder et al., 2018; Price et al., 2018).

281: Kandler et al. used mostly backscatter electron, except for small particles on TEM grids. Check also the other references please.
Corrected, and all the references were checked and updated

284: Is it possible to quantify the undercounting of backscatter versus secondary electron? That might be valuable information for people dealing with similar questions.
We thought about it, however, this would be extremely dependent on the aerosol sample so we decided to state it in a qualitative way.

283-298: Can you include an image showing the benefit of the Ir coating and the potential size increase? Again that appears to be valuable information.
The only thing we could do is taking some carbon coated images of some areas and some Ir coated images of different areas of the same filter (we cannot take Carbon coated images of an area, recoat it with Ir and go to the same area to take more images). The comparison would also be dependent upon the specific settings of the

instrument, hence we feel that a qualitative statement that we found Ir to be a better coating material is warranted, but a more detailed comparison is not.

301-304: it appears to be more meaningful to specify the pixel size in nm (scanning grid size), instead of the magnifications, which are screen-related.
Done

310: Was the ECD converted into aerodynamic or optical equivalent diameter or just used "as it is"? Please discuss, as this might introduce certain biases.
Added in third paragraph of Sect. 4: "This equivalent circular diameter hasn't been corrected or transformed into an optical or other equivalent diameter"
Added to fourth paragraph of Sect. 5: "Disagreement in the measurements can be also produced by the fact that the techniques are measuring different diameters; (optical and geometric)"

316: "evenly": In Fig. 7, a min/max variation of a factor of three is visible, interestingly without a size bias. Was it the same in all radial directions, or is that random fluctuations?
Added to fourth paragraph of Sect. 4: "In Fig. 7 one can see the radial distribution of aerosol particles on top a filter collected using the inlet system. In spite of some fluctuations (which are up to a factor 3 and appear to be random), one can see that the particles are homogenously distributed all over the central ~30mm of the filter. The areas were chosen by the user from all over the surface of the selected fraction of the filter".

320: Please 'link' "ECD" to "equivalent circular diameter".
Done

322-323: Where these charging problems observed despite the relative thick Ir coating?
Added to fifth paragraph of Sect. 4: "This reduces the likelihood of image defocusing over the SEM automated run".

We observed a frequent image focusing problem during long overnight runs, when the filters we were scanning had large numbers (above 20) of particles per image. We performed some tests and long exposure to the electron beam seemed to be the only reason of this defocusing effect. After adding the 12-15 particle limitation, this defocusing as a consequence of beam exposure effect was mostly eliminated.

414-415: I suggest treating this more precisely. At 10 µm, there is a disagreement of about a factor of 10 or slightly more, and the theory predicts between 2 and 5. Considering the uncertainties, it's probably fair to call this agreement. At 2 µm, there is the same factor > 10 difference, but the theory says 1. Here, 'agreement' becomes stretched. However, the optical particle counter curves have persistent minima (3 µm, 10 µm) and maxima (2 µm, 5 µm), where the SEM curve is smooth. Are these minima/maxima realistic or potentially an artifact of a failing Mie inversion?

We would rather keep this discussion qualitative for several reasons, not least that we are contrasting optical sizes with geometric sizes and also that the flow rates in the theoretical calculations were not identical to those in these specific experiments. However, the qualitative conclusion that the bypass being open reduces the isokinetic enhancement is valid and this should be written more clearly. We have amended the line to read:

(Added to second paragraph of Sect. 5) "The results of these comparisons are in qualitative agreement with the theoretical calculations in Sect. 2.2, i.e. that the sub-isokinetic enhancement is reduced with the bypass open."

In addition we have stated why we do not make a quantitative comparison or use the theory to 'correct' the data:

Added to end of Sect. 5 "Given the uncertainties on both techniques and the fact that they measure different diameters (optical diameter in the case of the PCASP-CDP and geometric equivalent circular diameter in the case of the SEM), this comparisons cannot be used to quantify the biases in the system, but can be used to make a qualitative comparison. For similar reasons, the SEM data hasn't been corrected using the theoretical efficiency."

In addition, on reviewing section 5 in light of the referee's comments we decided to restructure it. We now present the information in a more logical manner, which reflects the order of the figures. Please refer to the revised section 5 for the changes.

479: "sulphate aerosol particles, which are solid or liquid sulphuric acid particles" If it is sulphate (probable), then it is a salt as reaction product from an acid with something else. Solid sulphuric acid on a filter is improbable. Please correct the phrasing. Also, particles in this category could be organo-sulphates.

Yes, this description was poor. We have replaced this text with the following (Appendix B3):
"Aerosol particles in this category contained a substantial amount of S. This S might be in the form of inorganic or organic sulphate compounds. Some sulphate compounds, such as sulphuric acid, are relatively volatile and will be lost in the SEM chamber."

501: "chloride". Potassium-rich Cl- (and/or S-) containing particles are known from biomass burning (Li et al., 2003; Lieke et al., 2011), and other Cl-rich from (waste) incineration (Willison et al., 1989; Graedel and Keene, 1995).
Added to Sect 7.5

512: How about fractionated crystallization of a sea-water droplet on the filter, leading to separate NaCl, MgCl, CaCO3 or CaSO4 particles?
Added to Appendix B6: "Some Ca rich particles could originate from the crystallization of sea water, loosely attached to NaCl. The latter component would dominate over the rest of the elements of the conglomerate and they would appear as Na rich particles, unless they shatter in the air (Parungo et al., 1986){Andreae, 1986 #503}(Hoornaert et al., 1996)"

648-650: This approach appears to be questionable, as turbulence for an increasing diameter tube probably has an additional generation mechanism (inertia), compared to turbulence in a constant diameter tube (mostly by shear). Please comment.
Added to appendix A: "This approach doesn't account for potential additional inertial losses that could occur as a consequence of the enlargement of the flow in the conical section."

However, the angle of enlargement is small (5.7°). It was designed to be below 7° in order to avoid flow separation (Andreae et al., 1988). In addition, the bending towards the wall that the particles could experience as a consequence of this 5.7° expansion is smaller than the bending towards the wall that the particles already have before entering the nozzle because of the sub-isokinetic expansion of the flow, which has already been quantified.

654-669: The bend approximation assumes a smooth tube, too. If it was used for the droplet separator, the conditions are not met. Also, if the flow is decelerated during the bend, large particles might become accumulated on the outer side, which is not accounted for by the simple approximation. Please discuss.
In Brockmann (2011) (page 94) they suggest to use this approach for flow constrictions such as a tee.

Add line: "This assumption might underestimate the losses since some large aerosol particles will become accumulated in the bypass".

683: For diffusion a constant diameter bend can probably be well-approximated by a straight tube.
That is what we did. This is stated in Appendix A (9[th] paragraph).

691-703: While it is correct that the particles are retained by the filter, not necessarily all particle sizes can be analyzed by microscopy techniques (representatively), as the smaller particles might be deposited inside the pores, too.
Added in Appendix A (9[th] paragraph): "However, the fact that some aerosol particles with diameters below the pore size could be deposited in the filter pores and therefore not be detected by the SEM technique could contribute to the undercounting".

707: The referred equations apply to sharp-edged nozzles, while in the setup blunt and probably rounded ones are used (according to the aircraft engine inlet description). In particular the inlet rounding is done to mitigate misalignment effects (Hermann et al.,2001).

The criteria of the classification appears again in Belyaev and Levin (1974), where they state that inlets which had certain ratios in between the diameters of the inlet edge, thickness and angles could be considered thin-walled or thick-walled. According to them, the problem with the thick-walled nozzles is that the air streamlines are

distorted when they approach the inlet edge. Belyaev and Levin (1974) state that the ratio in between the external and the internal diameter of the inlet edge must be below 1.1, but we cannot really define this parameter because of the curved profile of the edge. An alternative criteria is that the ratio in between the thickness of the inlet edge and the diameter of the edge is below 0.05. Again, it is not possible to define the thickness of the edge. The numerical criteria thin/thick walled seems to be designed for truncated conical sections, not for curved edges like our case.

However, the inlet we are considered has been designed to "avoid distortion of the pressure field at the nozzle tip and the resulting problems associated with flow separation and turbulence" (Andreae et al., 1988), and it has been described as thin-walled in the literature (Talbot et al., 1990; Andreae et al., 2000; Formenti et al., 2003), because this design that avoids flow separation and turbulence places it closer to the "thin-walled" category than the "thick-walled" category. As a consequence, we decided to apply the thin-walled equations to it.

The fact that the experimental data shows the same trends in the inlet behaviour than predicted using the thin wall assumption helps to strengthen this assumption.

A short explanation of this has been added to the text (Fourth paragraph of Appendix A).

924: Caption "Polycarbonate". As many effects discussion above might be closer related to the volumetric flow rate than to the mass flow rate, it should be shown in addition. The Iceland/Cape Verde ratio is inverted for the two filter types or two inlet types. How can this be explained?
It is true that for the polycarbonate case, the Cape Verde sampling was consistently about 10 L min-1 above the Icelandic sampling. However, we don't believe there is enough Icelandic samples to say there is an inverse trend for the Teflon case.

**References**

Andreae, M.O., Berresheim, H., Andreae, T.W., Kritz, M.A., Bates, T.S. and Merrill, J.T. 1988. Vertical-Distribution of Dimethylsulfide, Sulfur-Dioxide, Aerosol Ions, and Radon over the Northeast Pacific-Ocean. *Journal of Atmospheric Chemistry.* **6**(1-2), pp.149-173.

Andreae, M.O., Elbert, W., Gabriel, R., Johnson, D.W., Osborne, S. and Wood, R. 2000. Soluble ion chemistry of the atmospheric aerosol and SO2 concentrations over the eastern North Atlantic during ACE-2. *Tellus B.* **52**(4), pp.1066-1087.

Belyaev, S.P. and Levin, L.M. 1974. Techniques for collection of representative aerosol samples. *Journal of Aerosol Science.* **5**(4), pp.325-338.

Brockmann, J.E. 2011. Aerosol Transport in Sampling Lines and Inlets. *Aerosol Measurement.* John Wiley & Sons, Inc., pp.68-105.

Chou, C., Formenti, P., Maille, M., Ausset, P., Helas, G., Harrison, M. and Osborne, S. 2008. Size distribution, shape, and composition of mineral dust aerosols collected during the African Monsoon Multidisciplinary Analysis Special Observation Period 0: Dust and Biomass-Burning Experiment field campaign in Niger, January 2006. *Journal of Geophysical Research Atmospheres.* **113**(D17), pp.1-17.

Formenti, P., Elbert, W., Maenhaut, W., Haywood, J. and Andreae, M.O. 2003. Chemical composition of mineral dust aerosol during the Saharan Dust Experiment (SHADE) airborne campaign in the Cape Verde region, September 2000. *Journal of Geophysical Research.* **108**, p8576.

Hoornaert, S., Van Malderen, H. and Van Grieken, R. 1996. Gypsum and Other Calcium-Rich Aerosol Particles above the North Sea. *Environmental Science & Technology.* **30**(5), pp.1515-1520.

Parungo, F.P., Nagamoto, C.T. and Harris, J.M. 1986. Temporal and spatial variations of marine aerosols over the Atlantic Ocean. *Atmospheric Research.* **20**(1), pp.23-37.

Price, H.C., Baustian, K.J., McQuaid, J.B., Blyth, A., Bower, K.N., Choularton, T., Cotton, R.J., Cui, Z., Field, P.R., Gallagher, M., Hawker, R., Merrington, A., Miltenberger, A., Neely Iii, R.R., Parker, S.T., Rosenberg, P.D., Taylor, J.W., Trembath, J., Vergara-Temprado, J., Whale, T.F., Wilson, T.W., Young, G. and Murray, B.J. 2018. Atmospheric Ice-Nucleating Particles in the Dusty Tropical Atlantic. *Journal of Geophysical Research: Atmospheres.* **123**(4), pp.2175-2193.

Rosenberg, P.D., Dean, A.R., Williams, P.I., Dorsey, J.R., Minikin, A., Pickering, M.A. and Petzold, A. 2012. Particle sizing calibration with refractive index correction for light scattering optical particle counters and impacts upon PCASP and CDP data collected during the Fennec campaign. *Atmospheric Measurement Techniques.* **5**(5), pp.1147-1163.

Ryder, C.L., Marenco, F., Brooke, J.K., Estelles, V., Cotton, R., Formenti, P., McQuaid, J.B., Price, H.C., Liu, D.T., Ausset, P., Rosenberg, P.D., Taylor, J.W., Choularton, T., Bower, K., Coe, H., Gallagher, M., Crosier, J., Lloyd, G., Highwood, E.J. and Murray, B.J. 2018. Coarse-mode mineral dust size distributions, composition and optical properties from AER-D aircraft measurements over the tropical eastern Atlantic. *Atmospheric Chemistry and Physics.* **18**(23), pp.17225-17257.

Seinfeld, J.H. and Pandis, S.N. 2006. *Atmospheric Chemistry and Physics: From Air Pollution to Climate Change.* Wiley.

Talbot, R.W., Andreae, M.O., Berresheim, H., Artaxo, P., Garstang, M., Harriss, R.C., Beecher, K.M. and Li, S.M. 1990. Aerosol Chemistry during the Wet Season in Central Amazonia - the Influence of Long-Range Transport. *Journal of Geophysical Research Atmospheres.* **95**(D10), pp.16955-16969.

Trembath, J. 2012. *Airborne CCN Measurements, Doctor of Philosophy*. thesis, University of Manchester.

von der Weiden, S.L., Drewnick, F. and Borrmann, S. 2009. Particle Loss Calculator – a new software tool for the assessment of the performance of aerosol inlet systems. *Atmospheric Measurement Techniques.* **2**(2), pp.479-494.

Young, G., Jones, H.M., Darbyshire, E., Baustian, K.J., McQuaid, J.B., Bower, K.N., Connolly, P.J., Gallagher, M.W. and Choularton, T.W. 2016. Size-segregated compositional analysis of aerosol particles collected in the European Arctic during the ACCACIA campaign. *Atmospheric Chemistry and Physics.* **16**(6), pp.4063-4079.

---

## Author Response (AR1)

**Author's response**

We would like to thank the referees for their comments. We attach the responses to both referees (blue) with their original comments (black). Below this, we attach the latest tracked version of the manuscript.

**Responses to Referee 1 (R1)**

The manuscript of Sanchez-Marroquin et al. deals with the characterization of an aircraft inlet frequently used in the British BAE-146 research aircraft. Despite of the importance of inlet characterization, it happens frequently that aerosol inlets are built and used, but remain uncharacterized. Therefore, these type of studies is valuable to rate the results of aerosol research done with the according systems, in particular with respect to their (size) representativity.

The authors compare an experimental approach for inlet transmission characterization with a theory-based one and come to the conclusion of a general approximate agreement. They propose a range of operational conditions based on their results.

The paper is mostly well-written; the methods are explained and applied. Some unclear sections remain (detailed below). References are adequate. However, some effort should be placed into the thermodynamic considerations, and the SEM part should be structured partly into a second publication. Also, some intentions for future work are given scattered through the paper, which should be either moved to the motivation section or omitted.

Thank you very much for this comments. They are very useful and will definitely improve the manuscript. We address the specific comments below.

**General remarks**

The paper goes into details about aerosol flows, but the properties and values reported in the text should be treated with more precision. E.g., flow rates are reported in L per minute, but it is unclear, whether this means volumetric L at the outside conditions, volumetric L at the inlet conditions, mass equivalent L at standard conditions. Line Added in Sect. 2.1: "The air flow through the filter (filter flow) is measured by a mass flow meter, which measures the sampled air mass and reports it in equivalent litres at standard conditions (273.15 k, 1013.529 hPa)". Line Added in Sect. 2.2: (volumetric L at standard conditions: 273.15 k, 1013.529 hPa).

Line Added in Sect. 2.2: (all the flow rates of our calculations are given in L min-1 at standard conditions: 273.15 k, 1013.529 hPa).

636-637: "All calculation were done under standard conditions" – Why? Most aerosol/carrier gas interactions depend on the air viscosity and free mean path, some on Reynolds number and therefore density. As result, most efficiency functions at the end have temperature and density in them. It doesn't seem to be a wise choice to neglect these dependencies, in particular not for aircraft measurements with their strong variation. Also, the thermodynamic conditions change considerably from the outside conditions through in inlet and tubing to the filter. Estimates were done using 'classical' aerosol aspiration / transmission formulas, which don't appear to be relevant in all cases (see detailed comments). A major question in this context is why the authors decided not to use computational fluid dynamics modelling. While these techniques are work-intensive and in turbulent situations also not necessarily precise, in particular for the inlet, diffusor and bend / inertial separator section, they might have been more useful.

**C2**

A temperature and pressure dependence test was performed. In the figure below, one can see the total efficiency for the 40L/min case (including the diffusion) for standard conditions, International standard atmosphere conditions at 0 and 3000m (the range in which the filter inlet system works. The differences are negligible. A similar negligible dependence was calculated for another inlet on board of the FAAM aircraft in Trembath (2012).

Line added at the third paragraph of Appendix A: "The effect of changes in pressure and temperature (and therefore air density and dynamic viscosity) that normally occur in the filter inlet system sampling range (0 to 3000 m) are negligible in all the used equations".

Further thermodynamic analysis in order to estimate the heating losses are impossible to carry out because of the lack of temperature and pressure measurement instruments through the inlet, which are not possible to have because of certification issues. This is a frequent problem of aircraft research. This is now mentioned in section:

Added in fourth paragraph of Sect. 5: "Also, volatilization of certain type of aerosol particles (which are more abundant in the submicron fraction (Seinfeld and Pandis, 2006)) can happen during heating (in this case produced by deceleration of the flow in the inlet) or sampling..."

The authors are aware that CFD (if carried out properly) would be a good way to characterise the inlet system, but decided not to include any CFD because it is far beyond the scope of the PhD project of the main author and it is very unlikely that it would change the conclusions of the paper. In addition, we think that the use of the appropriate empirical equations in combination with the comparison with underwing optical probes is an effective means of determining the best method of using the inlet and what biases can be expected.

The section 8 and 9 appear as a misfit in the context of this technical paper. I suggest removing it here and extending it into a standalone paper or letter in another journal. A proof of capability of SEM and measurements with the filter system on board the aircraft is not really required, as this has been done during decades (Johnson et al., 1991; Formenti et al., 2003; Chou et al., 2008; Johnson et al., 2012). In addition, the plots are shown, but remain un-discussed, and no context (e.g., meteorology, trajectories, campaign aims) is given. If anything of the SEM compositional results should be included, I suggest including the sensitivity tests for the classification scheme as function of the detection limit (currently in the supplement), as from this you can derive recommendations with respect to element quantification settings.

We interpreted that this comments refers to the sections 7 and 8 (SEM compositional categories, and an example of an application).

Sensitivity tests have now been included in the main paper (Appendix C).

Sect. 7 (The Section regarding to the compositional categories) has been moved to the Appendix B, in order to make the paper flow better.

Sect. 8 (now Sect 6), which includes some examples of what the technique can do has been kept just as an example without further discussion since this is a techniques paper. Further publications including all the SEM data collected by the authors and its discussions are already in preparation, and they will refer to this publication, rather than describing the technique in multiples SI sections of these future publications. The SEM technique has been used by other parallel projects in ground collected samples, which will also refer to this work.

Add in Sect. 8: "The purpose of this section is purely to give examples of the capabilities of the technique, further analysis is planned for subsequent papers"

On the comment about the SEM technique being used in the past. Yes, it has been used in the past, but our approach draws on elements of a number of previous studies and the classification scheme is novel. It therefore needs to be described somewhere. Hence, we think this techniques paper is the perfect place to include this.

**Detailed remarks / corrections**

Abstract 28-30: While this is surely true, it's not part of the paper. Removed

40 Missing "." Corrected

**55 "... has been limited." was not carried out?**

Some efforts have been made, we reviewed these previously in Price et al. (2018), but they are very limited. We are more specific in section 8 where we refer to the relevant papers. We have added the relevant references to this statement in the introduction as well.

57-67: These lines are more a summary than an introduction. As it is partly redundant to section 9, I suggest removing it.

Yes, this paragraph is the summary of what will come in the paper. We usually structure our papers in this way since it helps focus in on the specific objectives of the paper.

88: It doesn't get clear from the picture: does the 0.7 cm inlet have the inner edges rounded? From the references literature I would think that it is.

The word curved has been added to the text, as done in the given references.

97: The numbers indicate a high precision, which is usually not achieved by mass flow meters (1 - 3 % uncertainty). How water vapour was treated, which influences the reading?

The uncertainty of the MFM has been added (See caption of Fig. 4). Since the error is 1 % of the full scale (and this one is 400 L/min), the errors are above 1%.

Water vapour was neglected, since its effect is negligible. The difference in the heat capacity of dry air and saturated air at 20  $^{\circ}$ C is about 2%. The difference in the molar mass of dry air and saturated air at 25  $^{\circ}$ C is about 1.2%.

Add in Sect. 2.2: "The presence of water vapour hasn't been corrected since its effect is negligible."

**98: It probably reports the gas mass, not volume.**

It measures the gas mass but reports the equivalent volume at standard conditions.

101: Rietschle? Added in Sect. 2.1: Elmo Rietschle (Gardner Denver Inc.)

110: Particles are not necessarily lost (to the wall), but can be diluted (i.e. not entering the inlet). It is not clear how aerosol would be diluted in this inlet.

112-113: and depending on pressure and temperature...?

Added to second paragraph of Sect. 2.2: "The sampling efficiency of any inlet depends on the flow rates, and the flow regime (laminar vs turbulent), the pressure and the temperature."

122: It seems that to rate the importance of a mechanism, its effect needs to be compared to all others. Was this done, or were the only most probably important mechanisms selected? Please explain the reasoning.

We have included and excluded the same mechanisms as described in von der Weiden et al. (2009) (the reference has been added to this line in the text).

Add in third paragraph of Sect. 2.2: (a discussion on the choice of equations, how they have been applied and the excluded mechanisms can be found in Appendix A)

Added in last paragraph of Appendix A: "Other losses: Some mechanisms (thermophoresis, diffusiophoresis, interception, coagulation and re-entrainment of deposited particles) have not been considered, since they are second order mechanisms under our conditions when compared with the calculated mechanisms (Brockmann, 2011; von der Weiden et al., 2009) and for one of them (electrostatic deposition) it is not possible to quantify them. Electrostatic deposition is normally avoided by using grounded conductive materials so no electrical field exists within the tubing (Brockmann, 2011). Since the FAAM BAe-146 research aircraft is not grounded during the flight, we cannot state this mechanism is irrelevant. However, the experimental agreement between the SEM and optical probes suggest that this is a minor loss mechanism."

126-162: Too redundant with the appendix. Suggestion: either refer to the appendix and remove all short explanations here, or include the full discussion currently in the appendix. For a technical journal, also the latter would be appropriate.

Most of the explanations have been removed.

181-182: How does the bypass change the temperature in the inlet (probably mitigate heating by less deceleration)? Is the effect strong enough to impact on volatile particles?

The fact that it is not possible at all to have temperature and pressure measurements through the inlet system (for certification reasons) limits our understanding of the bypass system. We can only state qualitatively that, as you mention, they bypass will decrease heating trough less deceleration (and maybe removing some heat from the system).

213: It is somewhat surprising that the filter flow appears to be unregulated. Maybe, a regulation system should be included in future as well.

Yes, we are recommending this as a part of a mid-life upgrade of the FAAM aircraft.

217: "microscope" Fixed

244: "highly unlikely" instead of "not likely"? Added 262: "regarded" instead of "shown"? Added

263: Regarding the "reference": just recently, there was a publication showing size distribution distortion for the 'free-stream' instruments, too, (Spanu et al., 2019), which might be worth checking.

Although we are aware that the probes might have some sampling biases, as stated in Rosenberg et al. (2012), we still decided to use them as a reference, as in previous works (Chou et al., 2008; Young et al., 2016; Ryder et al., 2018; Price et al., 2018).

281: Kandler et al. used mostly backscatter electron, except for small particles on TEM grids. Check also the other references please.

Corrected, and all the references were checked and updated

284: Is it possible to quantify the undercounting of backscatter versus secondary electron? That might be valuable information for people dealing with similar questions.

We thought about it, however, this would be extremely dependent on the aerosol sample so we decided to state it in a qualitative way.

283-298: Can you include an image showing the benefit of the Ir coating and the potential size increase? Again that appears to be valuable information.

The only thing we could do is taking some carbon coated images of some areas and some Ir coated images of different areas of the same filter (we cannot take Carbon coated images of an area, recoat it with Ir and go to the same area to take more images). The comparison would also be dependent upon the specific settings of the

instrument, hence we feel that a qualitative statement that we found Ir to be a better coating material is warranted, but a more detailed comparison is not.

301-304: it appears to be more meaningful to specify the pixel size in nm (scanning grid size), instead of the magnifications, which are screen-related.

310: Was the ECD converted into aerodynamic or optical equivalent diameter or just used "as it is"? Please discuss, as this might introduce certain biases.

Added in third paragraph of Sect. 4: "This equivalent circular diameter hasn't been corrected or transformed into an optical or other equivalent diameter"

Added to fourth paragraph of Sect. 5: "Disagreement in the measurements can be also produced by the fact that the techniques are measuring different diameters; (optical and geometric)"

**316: "evenly": In Fig. 7, a min/max variation of a factor of three is visible, interestingly without a size bias. Was it the same in all radial directions, or is that random fluctuations?**

Added to fourth paragraph of Sect. 4: "In Fig. 7 one can see the radial distribution of aerosol particles on top a filter collected using the inlet system. In spite of some fluctuations (which are up to a factor 3 and appear to be random), one can see that the particles are homogenously distributed all over the central ~30mm of the filter. The areas were chosen by the user from all over the surface of the selected fraction of the filter".

320: Please 'link' "ECD" to "equivalent circular diameter".

**322-323: Where these charging problems observed despite the relative thick Ir coating?**

Added to fifth paragraph of Sect. 4: "This reduces the likelihood of image defocusing over the SEM automated run".

We observed a frequent image focusing problem during long overnight runs, when the filters we were scanning had large numbers (above 20) of particles per image. We performed some tests and long exposure to the electron beam seemed to be the only reason of this defocusing effect. After adding the 12-15 particle limitation, this defocusing as a consequence of beam exposure effect was mostly eliminated.

414-415: I suggest treating this more precisely. At 10  $\mu$ m, there is a disagreement of about a factor of 10 or slightly more, and the theory predicts between 2 and 5. Considering the uncertainties, it's probably fair to call this agreement. At 2  $\mu$ m, there is the same factor > 10 difference, but the theory says 1. Here, 'agreement' becomes stretched. However, the optical particle counter curves have persistent minima (3  $\mu$ m, 10  $\mu$ m) and maxima (2  $\mu$ m, 5  $\mu$ m), where the SEM curve is smooth. Are these minima/maxima realistic or potentially an artifact of a failing Mie inversion?

We would rather keep this discussion qualitative for several reasons, not least that we are contrasting optical sizes with geometric sizes and also that the flow rates in the theoretical calculations were not identical to those in these specific experiments. However, the qualitative conclusion that the bypass being open reduces the isokinetic enhancement is valid and this should be written more clearly. We have amended the line to read:

(Added to second paragraph of Sect. 5) "The results of these comparisons are in qualitative agreement with the theoretical calculations in Sect. 2.2, i.e. that the sub-isokinetic enhancement is reduced with the bypass open."

In addition we have stated why we do not make a quantitative comparison or use the theory to 'correct' the data:

Added to end of Sect. 5 "Given the uncertainties on both techniques and the fact that they measure different diameters (optical diameter in the case of the PCASP-CDP and geometric equivalent circular diameter in the case of the SEM), this comparisons cannot be used to quantify the biases in the system, but can be used to make a qualitative comparison. For similar reasons, the SEM data hasn't been corrected using the theoretical efficiency."

In addition, on reviewing section 5 in light of the referee's comments we decided to restructure it. We now present the information in a more logical manner, which reflects the order of the figures. Please refer to the revised section 5 for the changes.

479: "sulphate aerosol particles, which are solid or liquid sulphuric acid particles" If it is sulphate (probable), then it is a salt as reaction product from an acid with something else. Solid sulphuric acid on a filter is improbable. Please correct the phrasing. Also, particles in this category could be organo-sulphates.

Yes, this description was poor. We have replaced this text with the following (Appendix B3): "Aerosol particles in this category contained a substantial amount of S. This S might be in the form of inorganic or organic sulphate compounds. Some sulphate compounds, such as sulphuric acid, are relatively volatile and will be lost in the SEM chamber.'

501: "chloride". Potassium-rich Cl- (and/or S-) containing particles are known from biomass burning (Li et al., 2003; Lieke et al., 2011), and other Cl-rich from (waste) incineration (Willison et al., 1989; Graedel and Keene, 1995).

Added to Sect 7.5

**512: How about fractionated crystallization of a sea-water droplet on the filter, leading to separate NaCl, MgCl, CaCO3 or CaSO4 particles?**

Added to Appendix B6: "Some Ca rich particles could originate from the crystallization of sea water, loosely attached to NaCl. The latter component would dominate over the rest of the elements of the conglomerate and they would appear as Na rich particles, unless they shatter in the air (Parungo et al., 1986){Andreae, 1986 #503}(Hoornaert et al., 1996)"

648-650: This approach appears to be questionable, as turbulence for an increasing diameter tube probably has an additional generation mechanism (inertia), compared to turbulence in a constant diameter tube (mostly by shear). Please comment.

Added to appendix A: "This approach doesn't account for potential additional inertial losses that could occur as a consequence of the enlargement of the flow in the conical section."

However, the angle of enlargement is small (5.7°). It was designed to be below 7° in order to avoid flow separation (Andreae et al., 1988). In addition, the bending towards the wall that the particles could experience as a consequence of this 5.7° expansion is smaller than the bending towards the wall that the particles already have before entering the nozzle because of the sub-isokinetic expansion of the flow, which has already been quantified.

654-669: The bend approximation assumes a smooth tube, too. If it was used for the droplet separator, the conditions are not met. Also, if the flow is decelerated during the bend, large particles might become accumulated on the outer side, which is not accounted for by the simple approximation. Please discuss. In Brockmann (2011) (page 94) they suggest to use this approach for flow constrictions such as a tee.

Add line: "This assumption might underestimate the losses since some large aerosol particles will become accumulated in the bypass".

683: For diffusion a constant diameter bend can probably be well-approximated by a straight tube. That is what we did. This is stated in Appendix A (9th paragraph).

691-703: While it is correct that the particles are retained by the filter, not necessarily all particle sizes can be analyzed by microscopy techniques (representatively), as the smaller particles might be deposited inside the pores, too.

Added in Appendix A (9th paragraph): "However, the fact that some aerosol particles with diameters below the pore size could be deposited in the filter pores and therefore not be detected by the SEM technique could contribute to the undercounting'

707: The referred equations apply to sharp-edged nozzles, while in the setup blunt and probably rounded ones are used (according to the aircraft engine inlet description). In particular the inlet rounding is done to mitigate misalignment effects (Hermann et al., 2001).

The criteria of the classification appears again in Belyaev and Levin (1974), where they state that inlets which had certain ratios in between the diameters of the inlet edge, thickness and angles could be considered thin-walled or thick-walled. According to them, the problem with the thick-walled nozzles is that the air streamlines are

distorted when they approach the inlet edge. Belyaev and Levin (1974) state that the ratio in between the external and the internal diameter of the inlet edge must be below 1.1, but we cannot really define this parameter because of the curved profile of the edge. An alternative criteria is that the ratio in between the thickness of the inlet edge and the diameter of the edge is below 0.05. Again, it is not possible to define the thickness of the edge. The numerical criteria thin/thick walled seems to be designed for truncated conical sections, not for curved edges like our case.

However, the inlet we are considered has been designed to "avoid distortion of the pressure field at the nozzle tip and the resulting problems associated with flow separation and turbulence" (Andreae et al., 1988), and it has been described as thin-walled in the literature (Talbot et al., 1990; Andreae et al., 2000; Formenti et al., 2003), because this design that avoids flow separation and turbulence places it closer to the "thin-walled" category than the "thickwalled" category. As a consequence, we decided to apply the thin-walled equations to it.

The fact that the experimental data shows the same trends in the inlet behaviour than predicted using the thin wall assumption helps to strengthen this assumption.

**A short explanation of this has been added to the text (Fourth paragraph of Appendix A).**

924: Caption "Polycarbonate". As many effects discussion above might be closer related to the volumetric flow rate than to the mass flow rate, it should be shown in addition. The Iceland/Cape Verde ratio is inverted for the two filter types or two inlet types. How can this be explained?

It is true that for the polycarbonate case, the Cape Verde sampling was consistently about 10 L min-1 above the Icelandic sampling. However, we don't believe there is enough Icelandic samples to say there is an inverse trend for the Teflon case.

**References**

[revised manuscript text omitted]

**Responses to Referee 2 (R2)**

This paper presents a characterization of the filter inlet system of the research aircraft BAe146. It includes calculated inlet sampling and transmission efficiency, a description of the analysis of the filter samples by scanning electron microscopy (SEM), and a comparison of the size distributions obtained by SEM with underwing aerosol and cloud probes.

Unfortunately, the manuscript suffers from being vague at important points. Especially for a technical journal, a comparison between calculations and measurements needs to discussed in more detail. Also, expressions like "in agreement" are used frequently where a precise numbers (with error limits) would have been necessary. Thus, I cannot recommend publication in the current stage and suggest some major revision before publication.

**Major points:**

As said above, the manuscript lacks precise numbers. Many statements are vague, like "in agreement" or "minor fraction" etc. This is not sufficient for a technical journal.

We have amended Sect. 4 and in Sect. 5 in order to remove subjective statements where possible and replace them with more quantitative statements.

Furthermore, the SEM part is a description of the classification, but no further validation is done. Additional aircraft-based gas (e.g. CO) and particle measurements (mass spectrometers?) may help to characterize the air mass origin and the particle properties and thereby validate the composition.

We gave this some thought when planning these experiments, but concluded that it was not possible to find an established technique that we could quantitatively validate our SEM technique against. Validation of the size resolved composition would require significantly more detailed particle by particle information than could be inferred from tracers like CO or even from the available aerosol mass spectrometers. The AMS for example only provides information on the non-refractory components of fine mode aerosol.

In order to address the comment we have included a new figure (Fig 11) where we show an additional six size resolved compositions. This is accompanied by a new discussion in Sect. 6. Sect. 6 now focuses on examples and the paragraph on mass spec techniques has been removed. Fig. 7 includes data for SE England and Alaska (three samples for each). The extremely good agreement between the Figure 11c and 10a, which were samples from the same flight, helps to demonstrate the reproducibility. We also, show that the composition of the aerosol in the two locations is different in ways which we would expect, which shows that we are sensitive to different aerosol types and the composition varies in a consistent way.

Of course, the ideal situation would be to have a standard instrument to compare against, but given this standard instrument does not exist, we suggest that the best way forward would be to take part in a suitable inter-comparison at some point in the future.

The comparison of SEM size distribution with the PMS probes is not very conclusive, because only qualitative statements ("in good agreement") are made.

The discussion of the SEM-Optical probes (Sect. 5) comparisons has been restructured and improved.

Furthermore, the size distributions of the PCASP (Fig. 5) seem to have a problem at 300 nm and above 2 \_m. The PCASP shows decreasing number concentrations above 2 \_m while the CDP starts at 5 \_m with much higher number concentrations. Does the PCASP underestimate particle number above 2 \_m? If so, would it be better to omit these points and use a lognormal fit to the reliable CDP and PCASP data to obtain realistic fine and coarse mode distributions? To what extend can such size distributions validate the inlet efficiency if the uncertainties are so high?

Unfortunately there were some errors in Fig 5, which have now been corrected. Nevertheless, we sometime see an apparent discrepancy between the PCASP and CDP at above 2 um (the discrepancy at 300 nm is resolved using the correction detailed in Rosenberg et al. (2012) which is already discussed in the text (we accidently plotted the uncorrected data). In addition, horizontal error bars have been added to all the PCASP-CDP data.

In the reference paper for the PCAS\_CDP calibration, one can read: "Some bumps seen in the PCASP distribution have been accentuated by the calibration and refractive index correction presented here. It could be the case that these are real modes or there is the potential that this is an artefact caused by imperfect knowledge of the particle scattering properties" (Rosenberg et al., 2012). Hence, it is very difficult to address if the feature at 2 um is physical or just an artefact. However, in other data (Fig 8b, Fig 9a, b and c), these bumps cannot be seen as easily and in all the cases as in Fig 5 (which is the same data as Fig 8a), so they are likely to be physical.

Added to end of Sect. 5: "Some of the PCASP size distributions contain some bumps (above 2  $\mu$ m), but it is not possible to address if they are physical or just an artefact produced by the calibration (Rosenberg et al., 2012)."

We strongly disagree that it is a good idea to show fits for the comparisons instead of the data with errors. The fitting can have some subjective parameters (number of modes and restrictions on the fit) and not showing the actual data would potentially omit a lot of information. As a consequence we decided to keep only the data without any fitting on it and understand the uncertainties and potential artefacts of the system. In addition, Rosenberg et al. (2012) does not recommend showing a fit instead of the data as a way to deal with the bumps.

Figure 8-10: Have the SEM data been corrected for the calculated inlet transmission and aspiration efficiency? I could not find a statement on this in the text. If not, then an overestimation of about a factor 3 - 4 around  $10 \_m$  should be observed (from Fig 3b). Is that the case? By bare eye, the factor seems to be larger than three, but there is no discussion in the text, except for a "good agreement" statement.

We do not correct the data for the inlet efficiency. This is now clearly stated in the text (fifth paragraph of Sect 5):

"Given the uncertainties on both techniques and the fact that they measure different diameters (optical diameter in the case of the PCASP-CDP and geometric equivalent circular diameter in the case of the SEM), this comparisons cannot be used to quantify the biases in the system, but can be used to make a qualitative comparison. For similar reasons, the SEM data hasn't been corrected using the theoretical efficiency"

The referee refers to a factor of 3-4 enhancement. Based on the calculations we recommend that sampling is performed with total flow rates greater than 50 L min-1 with the bypass open, which result in enhancement smaller than about a factor of 2.

Regarding the 'good agreement' comment, we have made an effort to be more quantitative throughout the manuscript, particularly in discussion of the size distributions. We have reorganised section 5, also in light of the other referee's comments.

**Minor**

I was a bit confused by the mixture of sampling efficiency study and chemical composition study. I see that both needs to be done, but I needed some time to realize that the manuscript focuses on these two topics. Maybe a change of the title would help the reader.

Both aspects of the study are mentioned in the title, so it is not clear how we would change it to make it clearer. We reinforce this in the abstract and also in the (revised) final paragraph of the introduction.

**Specific comments**

Line 353: "This happens more frequently for smaller particles, but it can also happen with some larger particles..." What is "smaller" and "larger" here? Please be more precise and give a size range. Done.

Added to 6th paragraph of Sect 4.

Line 366-368: "The number of particles is very low, typically about the order of magnitude of one particle per 100 by 100 \_m square, which is well below the typical particle loading on a filter exposed to the atmosphere" Please give numbers for typical particle loading. "Well below" is not quantitative. Added to end of Sect. 4: "The number of particles is typically about the order of magnitude of one particle per 100 by 100  $\mu$ m square, which is more than an order of magnitude below all the samples in this study (apart from the sample shown in Fig 9c, where it is only about a factor 2)"

Line 373: "...from the analysis of atmospheric aerosol (it was only ever a very minor component)." Please specify "very minor".

Sentence was deleted for simplicity. The only purpose of the explanation was stating that that element is not very necessary for most of the aerosol studies.

Line 374: "By doing this, we make sure that we excluded more than half of the artefacts of the analysis" I don't understand. Before that, you said that >90% contained Cr, so you would remove >90 of the artefact, isn't it?

**Now it has been better explained:**

Added to end of Sect. 4: "In Fig. S2 one can see that about half of particles found in both blank filters and the handling blank belong to the metallic rich category. However, further examination of the composition of these metal rich particles revealed that almost all of them were Cr rich particles (about 97% in the case of the blank filters and about 96% in the case of the handling blank). As a consequence, we excluded all the Cr rich particles from the analysis of atmospheric aerosol. By doing this, we make sure that we exclude about half of the artefacts of the analysis"

**Section 7. Did you observe any signs of meteoric material (see Murphy et al., 2014)? Particles dominated by Fe, Mg, Si and S ?**

Although we did observe particles dominated by these elements, we cannot conclude that they are meteoric material since most of them were taken in the troposphere (most of them in the first kilometre), rather than the stratosphere where meteoric material has been observed. Analysis of meteoric material with the SEM seems more complicated since it only provides the weight percentages of the elements in the aerosol particles without any information about the isotope or the mass to charge ratio of what it is in the sample, but we will consider this while analysing the composition data which will be included in future papers.

**Line 501: "sodium chlorine" -> sodium chloride Done**

**Fig 4, caption: "FAAM core datasets" have not been explained before:**

Added to first paragraph of Sect. 3: "All the PCASP-CDP data shown here has been extracted from the FAAM cloud datasets corresponding to each specific flight via the Centre for Environmental Data Analysis"

Added to caption of Fig. 4: "The altitude data was extracted from the FAAM core datasets C019, C022, C024, C025, C058, C059, C060, C061, C062, C063, C085, C086, C087, C088, C089, C090 and C091 (via the Centre for Environmental Data Analysis)"

Fig 5 + lines 257-264: As already written above, the size distributions of the PCASP (Fig. 5) seem to have a problem at 300 nm and above 2  $\_$ m. The PCASP shows decreasing number concentrations above

2 \_m while the CDP starts at 5 \_m with much higher number concentrations. Does the PCASP underestimate particle number above 2 \_m? If so, would it be better to omit these points and use a lognormal fit to the reliable CDP and PCASP data to obtain realistic fine and coarse mode distributions? What happens at 10 \_m with the CDP?

We addressed the first points above.

In most cases, the CDP counting decreases around 10 um, but this is likely to be an actual measurement and not an artefact since particles above those sizes are relatively rare in the atmosphere.

Figs 8 and 9: I suggest combining Figs 8 and 9 into one figure with 4 graphs. Done

Fig 8, 9, 10 and line 415: "The results of these comparisons are in agreement with the theoretical calculations in Sect. 2.2." Did you correct the SEM size distribution with the calculated sampling efficiency? Can you divide SEM dN / PMS dN and derive an "experimental" sampling efficiency and compare that to the calculated curves in Sect. 2.2? One of the above should be done, otherwise your statement "are in agreement" is too weak.

We regard the efficiency calculations as qualitative, i.e. they provide a qualitative indication of losses and how to best use the inlet while minimising sampling biases. We therefore cannot use them to 'correct' the data, doing so would likely introduce a unquantifiable error to the data.

Added to end of Sect. 5: "Given the uncertainties on both techniques and the fact that they measure different diameters (optical diameter in the case of the PCASP-CDP and geometric equivalent circular diameter in the case of the SEM), this comparisons cannot be used to exactly quantify the biases on the system but understand its presence. For similar reasons, the SEM data hasn't been corrected using the theoretical efficiency"

[revised manuscript text omitted]
, 2005 #316@@author-year}\_{Kandler, 2007 314 #443@@author-year},\_{Chou, 2008 #447@@author-year},\_{Kandler, 2011 #442@@author-year},\_ 315 {Young, 2016 #75@@author-year},\_\_{Price, 2018 #450@@author-year}\_\_\_and\_{Ryder, 2018 #541@@author-year}. We use a Tescan VEGA3 XM scanning electron microscope (SEM) fitted with an 316 317 X-max 150 SDD Energy-Dispersive X-ray Spectroscopy (EDS) system controlled by an Aztec 3.3 318 software by Oxford Instruments, at the Leeds Electron Microscopy and Spectroscopy Centre (LEMAS) 319 at the University of Leeds. In order to get data from thousands of particles in an efficient way, data 320 collection was controlled by the AztecFeature software expansion.

321 Aerosol particles were collected with the filter inlet of the FAAM aircraft on polycarbonate track 322 etched filters with 0.4 µm pores (Whatman, Nucleopore). Samples for SEM are usually coated with 323 conductive materials in order to prevent the accumulation of charging on the sample surface {Egerton, 324 2005 #426}. For aerosol studies, materials like gold {Hand, 2010 #444}, platinum {Chou, 2008 #447}, 325 or evaporated carbon {Reid, 2003 #445;Krejci, 2005 #316;Young, 2016 #75} have been used. When it 326 comes to choosing which signal to detect, some previous studies used mainly backscattered electrons 327 {Reid, 2003 #445}{Kandler, 2007 #443}{Gao, 2007 #451}{Kandler, 2011 #442}{Young, 2016 #75;Price, 328 2018 #450}{Kandler, 2018 #602} and some others choose secondary electrons {Krejci, 2004 329 #316 [Hamacher-Barth, 2013 #500]. 
[revised manuscript text omitted]
 exactly quantify the biases on the system but understand its presence. For similar reasons, the SEM data hasn't been corrected using the theoretical efficiency. ¶ The results of these comparisons can be seen in Fig. 88 and Fig. 9 for both number and surface area size distribution. There are some discrepancies between the optical probes and the SEM size distributions from the filters, which has also been reported in previous works {Chou, 2008 #447}, {Young, 2016 #75; Price, 2018 #450}{Ryder, 2018 #541}. The SEM detected less submicron aerosol particles on the filter

**Deleted: 8 and Fig. 9**

|                              | Deleted: For the case shown in Fig 8b, it seems that the optical probes measured the same amount of aerosol abov      |  |  |  |  |  |  |
|------------------------------|------------------------------------------------------------------------------------------------------------------------------|--|--|--|--|--|--|
|                              | Deleted: a similar                                                                                                           |  |  |  |  |  |  |
|                              | Deleted: two                                                                                                                 |  |  |  |  |  |  |
|                              | Deleted: under                                                                                                               |  |  |  |  |  |  |
|                              | Deleted: bypass                                                                                                              |  |  |  |  |  |  |
|                              | Deleted: conditions                                                                                                          |  |  |  |  |  |  |
|                              | Deleted: suggest that the filter inlet system has the same type of bias for coarse aerosol particles as theoretically |  |  |  |  |  |  |
| $\langle \rangle$            | Deleted: again undercounting of submicron particles (even tough in this case it only happens for sizes bellow 400 nm) |  |  |  |  |  |  |
|                              | Deleted: Some undercounting of submicron aerosol can be seen below 500 nm.                                            |  |  |  |  |  |  |
| $\overline{\langle}$         | Deleted: In this case, both the optical probes and the SEM detected a very low amount of coarse aerosol particles            |  |  |  |  |  |  |
|                              | Deleted: counting                                                                                                            |  |  |  |  |  |  |
|                              | Deleted: )                                                                                                                   |  |  |  |  |  |  |
|                              | Deleted: This comparison does n'ot allow us to see any coarse mode enhancement because of the low coarse              |  |  |  |  |  |  |
| $\left\langle \right\rangle$ | Deleted: ), but one can see again the increasing undercounting of submicron aerosol particles (reaching an            |  |  |  |  |  |  |
| ())                          | Deleted: Overall,                                                                                                            |  |  |  |  |  |  |
| ()))                         | Deleted: d                                                                                                                   |  |  |  |  |  |  |
|                              | Deleted: of the submicron aerosol particles                                                                                  |  |  |  |  |  |  |
|                              | Deleted: The                                                                                                                 |  |  |  |  |  |  |
| $\backslash \rangle$         | Deleted: for the second comparison as one can see in Fig. 8b                                                          |  |  |  |  |  |  |
| $\langle \rangle$            | Deleted: very                                                                                                                |  |  |  |  |  |  |
| //                           | Deleted: Instead                                                                                                             |  |  |  |  |  |  |
|                              | Deleted: hannen                                                                                                              |  |  |  |  |  |  |

samples are exposed to high vacuum during the SEM analysis. In addition, the SEM techniques
 measure the dry diameter and the optical probes measure the aerosol diameter at ambient humidity.
 This hygroscopic effect shifts the dry size distributions to smaller sizes, which might also explain part
 of the disagreement {Nessler, 2003 #538;Young, 2016 #75}. Disagreement in the measurements can
 also be produced by the fact that the techniques are measuring different diameters (optical and
 geometric).

Some of the PCASP size distributions contain some 'bumps' (particularly above 2 µm), but it is not
possible to address if they are physical or just an artefact produced by the refractive index correction
{Rosenberg, 2012 #456}. Given the uncertainties on both techniques and the fact that they measure
different diameters (optical diameter in the case of the PCASP-CDP and geometric equivalent circular
diameter in the case of the SEM), this comparisons cannot be used to quantify the biases in the system,
but can be used to make a qualitative comparison. For similar reasons, the SEM data has not been
corrected using the theoretical efficiency.

661 662

663 6. Application to samples collected from the atmosphere above S.E. England and North Alaska

664 The SEM technique to produce size resolved composition of aerosol samples described in Sect. 4 has 665 been applied to samples collected from the FAAM aircraft in various locations. In Fig. 10 we show an 666 example of some of the capabilities of this technique applied to a sample collected in S. E. England, 667 The purpose of this section is purely to give examples of the capabilities of the technique, further 668 analysis is planned for subsequent papers. The fraction of particles corresponding to each 669 compositional category described in Appendix B for each size can be seen in Fig. 10a and the 670 corresponding number size distribution of each composition category can be seen in Fig. 10b. By 671 looking at this analysis, one can see that the sample carbonaceous aerosol particles made a substantial 672 contribution to the number across the full distribution and there was a clear mineral dust mode (Si 673 only, Si rich Al-Si rich and Ca rich) for particles larger than about 1 µm. There was also a smaller 674 contributions of metal rich and S rich aerosol particles, particularly in the fine mode. A potentially 675 useful application of the size resolved composition is calculating the surface area or mass of an 676 individual component of a heterogeneous aerosol. As an example, we have grouped the mineral dust 677 categories Si only, Si rich Al-Si rich and Ca rich to produce the surface area size distribution of mineral 678 dust (and potentially ash) in Fig. 10c.

679

680 In Fig. 11 we show six examples of the size-resolved composition of different aerosol samples in two 681 locations (South East England and North Alaska). We can see that the aerosol samples are very 682 different depending on the location. The aerosol samples collected in the UK shown in Fig 11a, c and 683 d are very similar to the sample shown in Fig. 10a. In fact the sample in Figure 10a was taken on the 684 same day in a similar location as the sample in Fig 11b and the similarity between the two helps to 685 demonstrate the reproducibility of our technique. Generally, these samples from S.E. England 686 contained carbonaceous aerosol throughout the size distribution, particularly in the fine mode. This 687 is consistent with typical urban influenced aerosol {Seinfeld, 2006 #473}. There is also a substantial 688 proportion of mineral dust and only a small proportion of Na rich aerosol. In contrast, the samples 689 collected in North Alaska (close or above the Arctic Ocean) generally contained a smaller proportion 690 of carbonaceous particles, but much larger contributions of Na rich aerosol (very likely sea salt 691 particles, since they were collected in a marine environment). The S-rich category was also substantial

**Deleted: be**

**Deleted: is**

**Deleted: '**

**Deleted: ¶**

In Fig. 10 we have presented some other SEM size distributions compared with the PCASP-CDP data from three different aerosol samples in contrasting locations. Since these data were taken during the scientific field campaigns and not test flights, we only collected one polycarbonate filter for SEM since the other line was used for INPs analysis on Teflon filters. All the sampling was done with the bypass open. The agreement between the optical and SEM obtain

| Moved down [2]: 6. Recommendations for aerosol |                                                    |   |  |  |  |  |
|------------------------------------------------|----------------------------------------------------|---|--|--|--|--|
| Deleted:                                       | ۹ (                                                |   |  |  |  |  |
| Moved d                                        | own [1]: 7. SEM compositional categories ¶         |   |  |  |  |  |
| Deleted:                                       | The data shown in this section have been obtain    |   |  |  |  |  |
| Deleted:                                       | rules described in the supplementary informatio    |   |  |  |  |  |
| Deleted:                                       | a                                                  |   |  |  |  |  |
| Deleted:                                       | Here                                               |   |  |  |  |  |
| Deleted:                                       | developed                                          |   |  |  |  |  |
| Deleted:                                       | one of the                                         |   |  |  |  |  |
| Deleted:                                       | S                                                  |   |  |  |  |  |
| Deleted:                                       | ; the resulting size resolved composition is show  | _ |  |  |  |  |
| Deleted:                                       | an                                                 |   |  |  |  |  |
| Deleted:                                       | so no further scientific analysis has been include | _ |  |  |  |  |
| Deleted:                                       | Sect. 7                                            |   |  |  |  |  |
| Deleted:                                       | . The                                              |   |  |  |  |  |
| Deleted:                                       | SEM                                                |   |  |  |  |  |
| Deleted:                                       | was clearly dominated by                           |   |  |  |  |  |
| Deleted:                                       | in all the sizes, but                              |   |  |  |  |  |
| Deleted:                                       | and some                                           |   |  |  |  |  |
| Deleted:                                       | other aerosol types (                              |   |  |  |  |  |
| Deleted:                                       | )                                                  |   |  |  |  |  |
| Deleted:                                       | There are very few ways to obtain the size-        | _ |  |  |  |  |
| Deleted:                                       | 6                                                  |   |  |  |  |  |
| Deleted:                                       | 0                                                  |   |  |  |  |  |
| Deleted:                                       | (Seinfeld and Pandis, 20)                          |   |  |  |  |  |
| Deleted:                                       | , which was previously described (the latter and   | _ |  |  |  |  |
| Deleted:                                       | exhibited much less                                |   |  |  |  |  |
| Deleted:                                       | taken                                              |   |  |  |  |  |

[revised manuscript text omitted]
 mechnanism,                                                                                                                                                                                                                                                                                                                                                                                          | is not possible to know the distribution of charge in the             |
| 1152                         |                                                                                                                                                                                                                                                                                                                                                                                                                 |
aerosol particles or the aircraft so it hasn't been included.     |
| 1152                         |                                                                                                                                                                                                                                                                                                                                                                                                                 | Deleted: ¶                                                            |
| 1153                         | Appendix B. SEM compositional categories                                                                                                                                                                                                                                                                                                                                                                        |
Moved (insertion) [1]                                             |
| 1154                         | Here we describe the 10 categories we have used in our compositional analysis, which are a summary                                                                                                                                                                                                                                                                                                              | Deleted: 7.                                                           |
| 1155                         | of the 32 rules described in the supplementary information. The approach has some similarities with                                                                                                                                                                                                                                                                                                             |                                                                       |
| 1156                         | the ones in previous studies {Krejci, 2005 #316;Chou, 2008 #447;Kandler, 2011 #442;Hand, 2010                                                                                                                                                                                                                                                                                                                   |                                                                       |
| 1157                         | #444;Young, 2016 #75}, but it is distinct. Because of the fact that the filter is made of C and O,                                                                                                                                                                                                                                                                                                              |                                                                       |
| 1158                         | background elements (C and O) were detected in all the particles. Particles in each category can                                                                                                                                                                                                                                                                                                                |                                                                       |
| 1159                         | contain smaller amounts of other elements apart from the specified ones. This classification scheme                                                                                                                                                                                                                                                                                                             |                                                                       |
| 1160                         | has been designed a posteriori to categorise the vast majority of the aerosol particles in the three field                                                                                                                                                                                                                                                                                                      |                                                                       |
| 1161                         | campaigns previously described and some ground collected samples in the UK and Barbados. The main                                                                                                                                                                                                                                                                                                               |                                                                       |
| 1162                         | limitation of the classification scheme is the difficulty to categorise internally mixed particles. The                                                                                                                                                                                                                                                                                                         |                                                                       |
| 1163                         | algorithm has been built in a way it can identify mixtures of mineral dust and sodium chloride (they                                                                                                                                                                                                                                                                                                            |                                                                       |
| 1164                         | appear as mineral dust but they could be split into a different category if necessary) and sulphate or                                                                                                                                                                                                                                                                                                          |                                                                       |
| 1165                         | nitrate ageing on sodium chloride (they appear as Na rich but it could also be split into a different                                                                                                                                                                                                                                                                                                           |                                                                       |
| 1166                         | category). However, other mixtures of aerosol wouldn't be identified, and they would be categorised                                                                                                                                                                                                                                                                                                             |                                                                       |
| 1167                         | by the main component in the internal mixture in most cases.                                                                                                                                                                                                                                                                                                                                                    |                                                                       |
| 1168                         | B.1. Carbonaceous                                                                                                                                                                                                                                                                                                                                                                                        |
| 1169                         | The particles in this category contained only background elements (C and O). The components of the                                                                                                                                                                                                                                                                                                              |                                                                       |
| 1170                         | carbonaceous particles consist in either black carbon from combustion processes or organic material,                                                                                                                                                                                                                                                                                                            |                                                                       |
| 1171                         | which can be either directly emitted from sources or produced by atmospheric reactions {Seinfeld,                                                                                                                                                                                                                                                                                                               |                                                                       |
| 1172                         | 2006 #473}. Particles containing certain amount of K and P in addition to the background elements                                                                                                                                                                                                                                                                                                               |                                                                       |
| 1173                         | were also accepted in these category. These elements are consistent with biogenic origin aerosol                                                                                                                                                                                                                                                                                                                |                                                                       |
| 1174                         | particles {Artaxo, 1995 #472}. Distinction between organic and black carbon aerosol unfortunately                                                                                                                                                                                                                                                                                                               |                                                                       |
| 1175                         | could not reliably be done. Since N is not analysed in our SEM set up, any nitrate aerosol particle would                                                                                                                                                                                                                                                                                                       |                                                                       |
| 1176                         | tail into this category if it is on the filter. However, since these particles are semi-volatile, some of                                                                                                                                                                                                                                                                                                       |                                                                       |
| 1170                         | these aerosol particles would not resist the low pressure of the SEM chamber. This could be further                                                                                                                                                                                                                                                                                                             |                                                                       |
| 11/8
8                    | nivesugateu in the luture.                                                                                                                                                                                                                                                                                                                                                                               |                                                                       |
| 1179                         | B.2. S rich                                                                                                                                                                                                                                                                                                                                                                                              |
| 1                            |                                                                                                                                                                                                                                                                                                                                                                                                                 |                                                                       |

| Aerosol particles in this category contained a substantial amount of S. This S might be in the form of     |              |                                                                                                                                                                                      |
|------------------------------------------------------------------------------------------------------------|--------------|--------------------------------------------------------------------------------------------------------------------------------------------------------------------------------------|
| inorganic or organic sulphate compounds. Some sulphate compounds, such as sulphuric acid, are              |              |                                                                                                                                                                                      |
| relatively volatile and will be lost in the SEM chamber                                                    |              | Deleted: Aerosol particles in this category contained a                                                                                                                       |
| B.3 Metal rich                                                                                      |              | substantial amount of S. These EDS signals are compatible
with sulphate aerosol particles, which are solid or liquid
culphuic acid particles (Kumar, 2017 #E32) in the same un |
| The composition of particles in this category is dominated by one of the following metals: Fe, Cu, Pb,     | $\backslash$ | as the nitrates, this particles are semi-volatile and some of